# Basal ganglia output reflects internally-specified movements

Mario J Lintz[1,2,3], Gidon Felsen[1,2,3]*

[1]Department of Physiology and Biophysics, University of Colorado School of Medicine, Aurora, United States; [2]Neuroscience Program, University of Colorado School of Medicine, Aurora, United States; [3]Medical Scientist Training Program, University of Colorado School of Medicine, Aurora, United States

**Abstract** How movements are selected is a fundamental question in systems neuroscience. While many studies have elucidated the sensorimotor transformations underlying stimulus-guided movements, less is known about how internal goals – critical drivers of goal-directed behavior – guide movements. The basal ganglia are known to bias movement selection according to value, one form of internal goal. Here, we examine whether other internal goals, in addition to value, also influence movements via the basal ganglia. We designed a novel task for mice that dissociated equally rewarded internally-specified and stimulus-guided movements, allowing us to test how each engaged the basal ganglia. We found that activity in the substantia nigra pars reticulata, a basal ganglia output, predictably differed preceding internally-specified and stimulus-guided movements. Incorporating these results into a simple model suggests that internally-specified movements may be facilitated relative to stimulus-guided movements by basal ganglia processing.

## Introduction

As we interact with the world, our movements are selected based on external sensory stimuli and internal variables representing action value, learned stimulus-response contingencies, and prior experiences (*Gold and Shadlen, 2007*). Selecting the movement associated with the most desirable outcome requires appropriately weighting each of these factors. While the neural substrates for movements based on external sensory stimuli have been the focus of much research (*Hall and Moschovakis, 2003*), where, how, and when internal goals influence movement selection is less well understood. The basal ganglia (BG) are known to be involved in motor control (*Hikosaka and Wurtz, 1989*; *Mink, 1996*), contributing to movement selection by modulating inhibition on competing downstream motor structures (*Basso and Wurtz, 2002*; *Di Chiara et al., 1979*; *Hikosaka and Wurtz, 1985*). In particular, the BG have been thought to bias the selection of movements towards those associated with the highest value (*Hikosaka et al., 2006*). This 'value-biasing' hypothesis is supported by much evidence showing that activity in several BG nuclei is modulated, prior to stimulus presentation, by reward expectation (*Bryden et al., 2011*; *Handel and Glimcher, 2000*; *Hikosaka et al., 2006*; *Kawagoe et al., 1998*; *Sato and Hikosaka, 2002*) such that movements toward high-value targets are disinhibited relative to movements toward low-value targets (*Hikosaka et al., 2006*; *Lauwereyns et al., 2002*). Anatomical evidence is consistent with a primary role for the BG in mediating the integration of value-based information into motor plans (*Bolam et al., 2000*; *Gerfen and Surmeier, 2011*). However, movement selection may also be guided by other internal representations, such as recent movements and their outcomes (*Corrado et al., 2005*; *Fecteau and Munoz, 2003*; *Lau and Glimcher, 2005*). We therefore asked whether BG activity mediates the influence of internal goals, in addition to value, on movement selection.

*For correspondence: gidon.felsen@gmail.com

Competing interests: The authors declare that no competing interests exist.

**eLife digest** An important role of the nervous system is to allow us to move around in the world. These movements are typically influenced by the goal that we want to achieve (for example, finding food) as well as stimuli that we sense in our environment (for example, the smell of pizza). Yet we understand little about how the brain controls these sorts of goal-directed movements, even under normal conditions. This lack of basic understanding presents a big problem when it comes to treating movement disorders like Parkinson's disease.

For a long time, a collection of brain regions called the basal ganglia have been known to be important for controlling movements, although the specific role that they play in this process is not well understood. Does the brain activity that controls movements differ depending on whether the movement is made in response to a stimulus or not?

Using mice, Lintz and Felsen have now recorded the activity of individual neurons in the basal ganglia that signal to other brain regions as the animals performed a behavioral task. Different trials in the task required the mouse to make two types of otherwise-identical movements: movements based on a stimulus, and movements based on recent experiences (and not triggered by a stimulus). The output activity of the basal ganglia differed under these two conditions, suggesting that the basal ganglia may play different roles in each type of movement.

From the results, Lintz and Felsen could make some predictions about how the basal ganglia influence the activity of downstream regions of the nervous system that control movement. Further studies are now required to test these predictions.

We reasoned that, if this were the case, BG output would differ when selecting equally valuable stimulus-guided and internally-specified movements. Specifically, we would expect that internally-specified movements would be promoted relative to otherwise-identical stimulus-guided movements, just as more valuable movements have been shown to be promoted relative to otherwise-identical less valuable movements (*Hikosaka et al., 2006*; *Sato and Hikosaka, 2002*). Notably, it is has been proposed that Parkinsonian patients exhibit more pronounced bradykinesia when initiating internally-specified than stimulus-guided movements because the latter engage pathways outside of the BG (*Glickstein and Stein, 1991*). However, whether the BG themselves are differentially engaged by these two types of movements has not been tested.

We distinguished between these two possibilities by recording from neurons in the substantia nigra pars reticulata (SNr), an output nucleus of the BG critical for orienting movements (*Basso and Sommer, 2011*; *Handel and Glimcher, 1999*; *Hikosaka and Wurtz, 1983a*), in mice performing a behavioral task in which a sensory stimulus either was or was not informative of the rewarded direction of an orienting movement. Using a design akin to that of other recent studies (*Pastor-Bernier and Cisek, 2011*; *Seo et al., 2012*; *Ito and Doya, 2015*), in alternating blocks of trials, the rewarded direction was either determined by a sensory cue or by internal representations informed by recent trial history. Critically, we designed the task such that correct movements were equally valuable in both conditions. We found that SNr activity predictably differed between these two conditions, supporting the idea that the BG mediate the influence on movement selection of internal goals. We interpret these results, in the context of a simple model of BG output (*Hikosaka et al., 2006*), as suggesting that internally-specified movements may be promoted over stimulus-guided movements by BG activity.

## Results

### Behavioral assay dissociates stimulus-guided and internally-specified movements

We trained mice on a delayed-response spatial choice task comprised of interleaved blocks of 'stimulus-guided' (SG) trials, in which the direction of movement is selected based on a sensory stimulus (*Uchida and Mainen, 2003*), and 'internally-specified' (IS) trials, in which the direction of an otherwise-identical movement is selected based on internal representations informed by recent trial

history (see Materials and methods; *Figure 1A,B*). In each trial of the task, the mouse is presented with a binary odor mixture at a central port, waits for an auditory go cue, and moves to the left or right reward port for a water reward. In SG trials, the dominant component of the odor mixture – which varies trial by trial – determines the side at which reward will be delivered, while in IS trials, a balanced mixture of the two odors is always presented but reward is delivered at only one side throughout the block (see Materials and methods; *Figure 1B*). Thus, while both trial types require the mouse to sample the stimulus, in SG trials the stimulus indicates that the rewarded side is determined by the odor mixture and in IS trials the stimulus indicates that the rewarded side is determined by the recent history of choices and outcomes. We found that mice were able to infer (unsignaled) transitions between the SG and IS blocks and switch their response mode accordingly: during SG blocks, mice were equally likely to choose the left and right port (*Figure 1C*, gray boxes) reflecting a dependence on the odor mixture (*Figure 1D*), while during IS blocks, mice reliably returned to the same (rewarded) port in each trial (*Figure 1C*, white boxes). We quantified performance in IS blocks by calculating, for each block, the percentage of correct trials and the number of error events, defined as a run of consecutive incorrect choices (*Figure 1E*). Finally, we reasoned that if a mouse were to recognize that a given trial belonged to an IS block, it could prepare its movement in advance and would therefore be able to reach the reward port faster (*Niemi and Näätänen, 1981*; *Poulton, 1950*; *Seo et al., 2012*). Indeed, across the population of sessions, we found that reaction time – defined as the time from the go cue to reward port entry – was shorter in IS trials than in the 'easy' SG trials in the same session [*Figure 1F*; we used only easy SG trials (mixture ratios of 95/5, 80/20, 20/80, and 5/95) to control for a potential dependence of reaction time on difficulty; population of sessions: p = $1.7 \times 10^{-11}$, paired t-test; individual sessions: IS shorter than SG in 46/108, SG shorter than IS in 3/108, p<0.05, Wilcoxon rank-sum test; ipsiversive and contraversive trials compared separately]. Together, these data suggest that, as intended, the direction of movement in SG blocks is selected based on the stimulus while the direction of movements in IS blocks is selected based on recent trial history. We therefore utilized this behavioral assay to compare how stimulus-guided and internally-specified movements are mediated by the BG, as described below.

## SNr activity differs for stimulus-guided and internally-specified movements

If the BG integrates not only value but also other internal representations, then stimulus-guided and internally-specified movements may differentially engage the BG despite being equally valuable. In this case, we would predict that BG output would depend on whether the movement was internally specified or stimulus guided, and specifically, in our task, on the degree to which recent trial history is informative of rewarded direction. To test this prediction we examined activity in the SNr, a BG output known to be involved in orienting movements (*Basso and Sommer, 2011*; *Handel and Glimcher, 2000*, *1999*; *Hikosaka and Wurtz, 1983a*, *1983b*). We recorded from 296 well-isolated left SNr neurons (see Materials and methods; *Figure 2*) in four mice performing the task (see *Supplementary file 1*). Data from one example neuron are shown in *Figure 3A,B*, segregated by reward port selected (ipsilateral vs. contralateral to the recording side) and by trial type (SG vs. IS). The activity of this neuron clearly depends on both movement direction and trial type. To examine these dependencies across the population of neurons we first examined firing rate during the delay epoch, defined as the time from odor valve open to the time of odor port exit (*Figure 1A*, gray box), which most directly captures, across trial types, activity underlying selection of the direction of movement (but since activity in IS trials may, by design, reflect direction selection even before stimulus delivery, we subsequently examine activity in other epochs). Based on the firing rate during this epoch in SG and IS trials, we then calculated direction preference (see Materials and methods). This value ranges from -1 (strongly 'prefers' ipsiversive) to 1 (strongly prefers contraversive), where 0 represents no preference. We found that 216/296 neurons displayed a significant direction preference (p<0.05) during the delay epoch, with about as many preferring ipsiversive (94/216) as contraversive (122/216) movements (*Figure 3C*). Since SNr activity has been shown to exhibit both movement-related increases and decreases (*Bryden et al., 2011*; *Gulley et al., 2002*, *1999*; *Handel and Glimcher, 1999*; *Sato and Hikosaka, 2002*), we next asked whether a relationship existed between direction preference and the sign of activity change during the delay epoch, relative to baseline (see Materials and methods). We found that 188/296 neurons exhibited an increase in activity during this epoch, 91/296 exhibited a decrease, and 17/296 exhibited no change (*Table 1*), consistent with

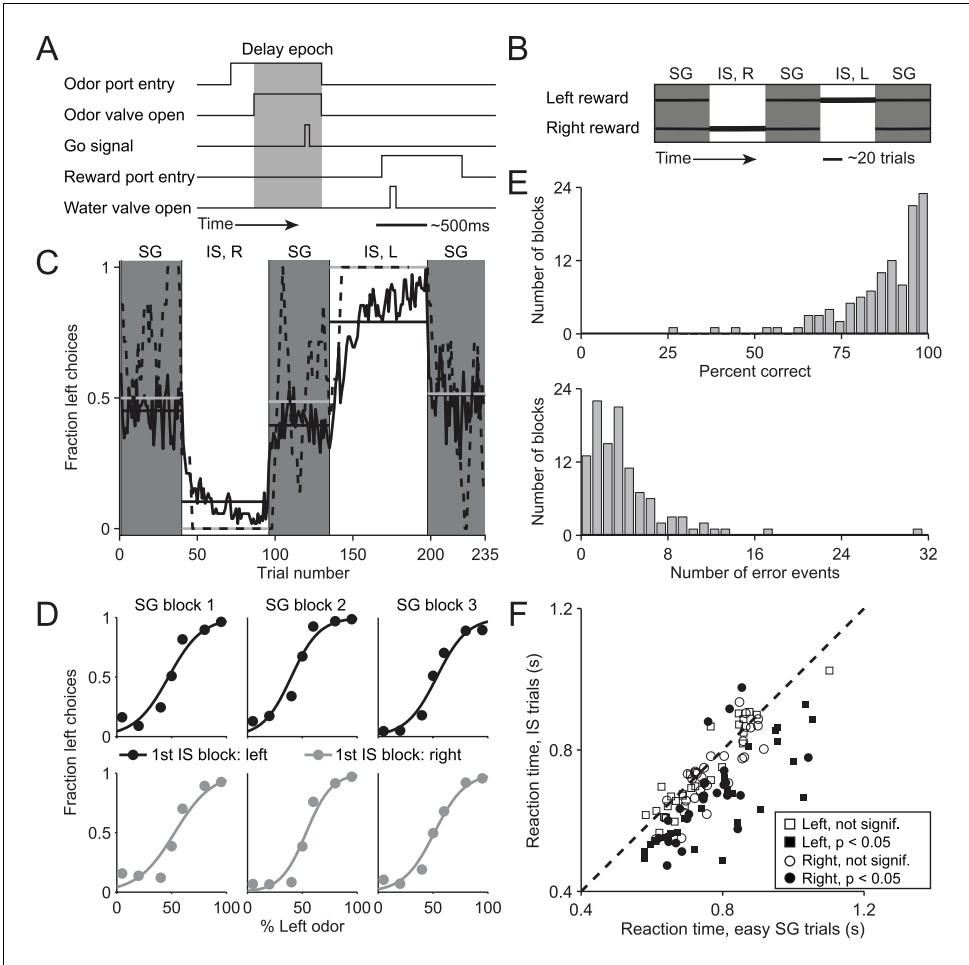

**Figure 1.** Behavioral task and performance. (**A**) Timing of events in each trial. The mouse enters the odor port, receives an odor mixture, waits for the go signal, exits the odor port, moves to one of the reward ports, and receives a water reward for a correct choice. Gray box, delay epoch. (**B**) Organization of SG (gray) and IS (white) blocks within a session. All sessions start with an SG block and alternate between SG and IS blocks. In SG blocks, reward side corresponds to the dominant odor in the mixture [(-)-carvone, left; (+)-carvone, right]; when the odors are balanced ([(-)-carvone] = [(+)-carvone]), the probability of reward at both reward ports is 0.5. In IS blocks, odors are balanced in every trial and reward is available at the same side in each trial. Thickness of horizontal lines corresponds to probability of reward. SG, stimulus guided; IS, internally specified; L, left; R, right. (**C**) Fraction of left choices across block types throughout the session. Dashed line shows an example session (boxcar smoothed over 7 trials), solid black line shows mean over all sessions (54, from 4 mice), horizontal black lines show block means, horizontal gray lines show ideal block means (if all choices were correct). To account for different numbers of trials per block across sessions, trials that occur in < 60% of sessions are excluded. In SG blocks only difficult trials [(+)-carvone/(-)-carvone = 60/40, 50/50, or 40/60] are shown. (**D**) Mean performance in SG blocks over all sessions, separated by rewarded side of first IS block in the session. Lines show best fit to $p = \frac{1}{1+e^{-a-bx}}$, where $x$ is the proportion of the left odor [(-)-carvone)] in the mixture, $p$ is the fraction of right choices, and $a$ and $b$ are free parameters. While choices were slightly biased by the rewarded direction in the first IS block (center panels), they were much more strongly influenced by the stimulus. (**E**) Performance in IS blocks. Histograms of percent correct choices (top) and number of error events (run of consecutive incorrect choices, bottom) across blocks over all sessions. (**F**) Mean reaction time in easy SG trials plotted against mean reaction time in IS trials in the corresponding session, separately for each direction of movement.

previous studies (*Bryden et al., 2011*; *Gulley et al., 2002*, *1999*; *Handel and Glimcher, 2000*, *1999*; *Sato and Hikosaka, 2002*). Within these groups, neurons exhibited a preference for ipsiversive, contraversive, or neither direction in roughly equal numbers (*Table 1*).

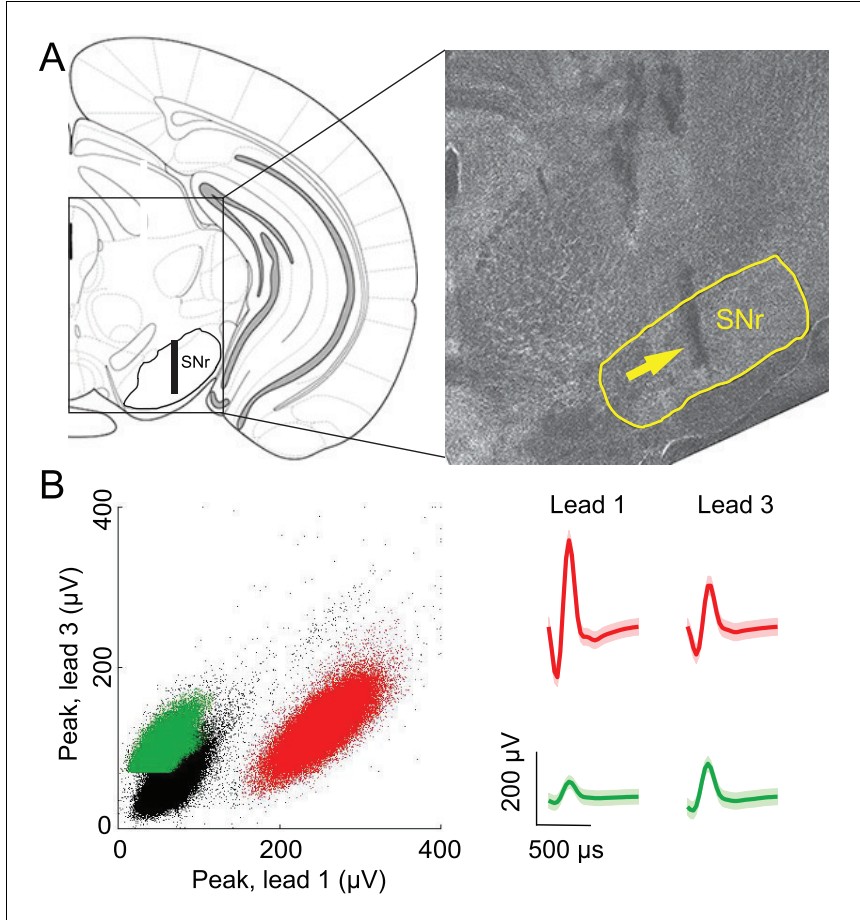

**Figure 2.** Confirmation of recording sites and spike clustering. (**A**) Schematic (left) shows targeted recording extent (bar) within SNr; coronal section (right, 3.3 mm caudal from bregma) shows representative tetrode track (arrow) in SNr. (**B**) Left, peaks of waveforms from lead 1 plotted against peaks of waveforms from lead 3 of one tetrode for a representative recording session. Note that clustering was performed using additional features to those shown here. Red and green points show waveform peaks recorded from neurons considered to be distinct. Right, waveforms (mean ± SD) corresponding to red and green points.

We next examined whether activity during the delay epoch of direction-selective neurons (*Figure 3C*, black bars) differed between SG and IS trials, in two complementary ways. First, we examined whether the difference in activity between ipsiversive and contraversive trials depended on whether the movement was stimulus guided or internally specified. Across our population, neurons tended to show a larger difference in firing rate preceding ipsiversive and contraversive movements in IS than in SG trials [*Figure 3D*; ipsiversive-preferring neurons: $p = 2.2 \times 10^{-10}$, paired t-test; contraversive-preferring neurons: $p = 4.0 \times 10^{-4}$, paired t-test]. Second, we determined whether neurons were trial-type-dependent by comparing firing rates between SG and IS trials in which the selected movement was correct and in the preferred direction of the neuron; we then repeated this comparison for the antipreferred direction. For trials in the preferred direction, we found that the activity of approximately half of the direction-selective neurons was modulated by trial type (101/216; p<0.05, unpaired t-test), with more neurons exhibiting higher activity in IS trials than SG trials (84/101 vs. 17/101; $p = 2.6 \times 10^{-11}$, $X^2$ test). Conversely, for trials in the antipreferred direction, we again found that the activity of approximately half of the direction-selective neurons was modulated by trial type (109/216; p<0.05, unpaired t-test; 158/216 direction-selective neurons were modulated by trial type in at least one direction), but that more neurons exhibited higher activity in SG trials than IS trials (76/109 vs. 33/109; $p = 3.8 \times 10^{-5}$, $X^2$ test). Therefore, while we found

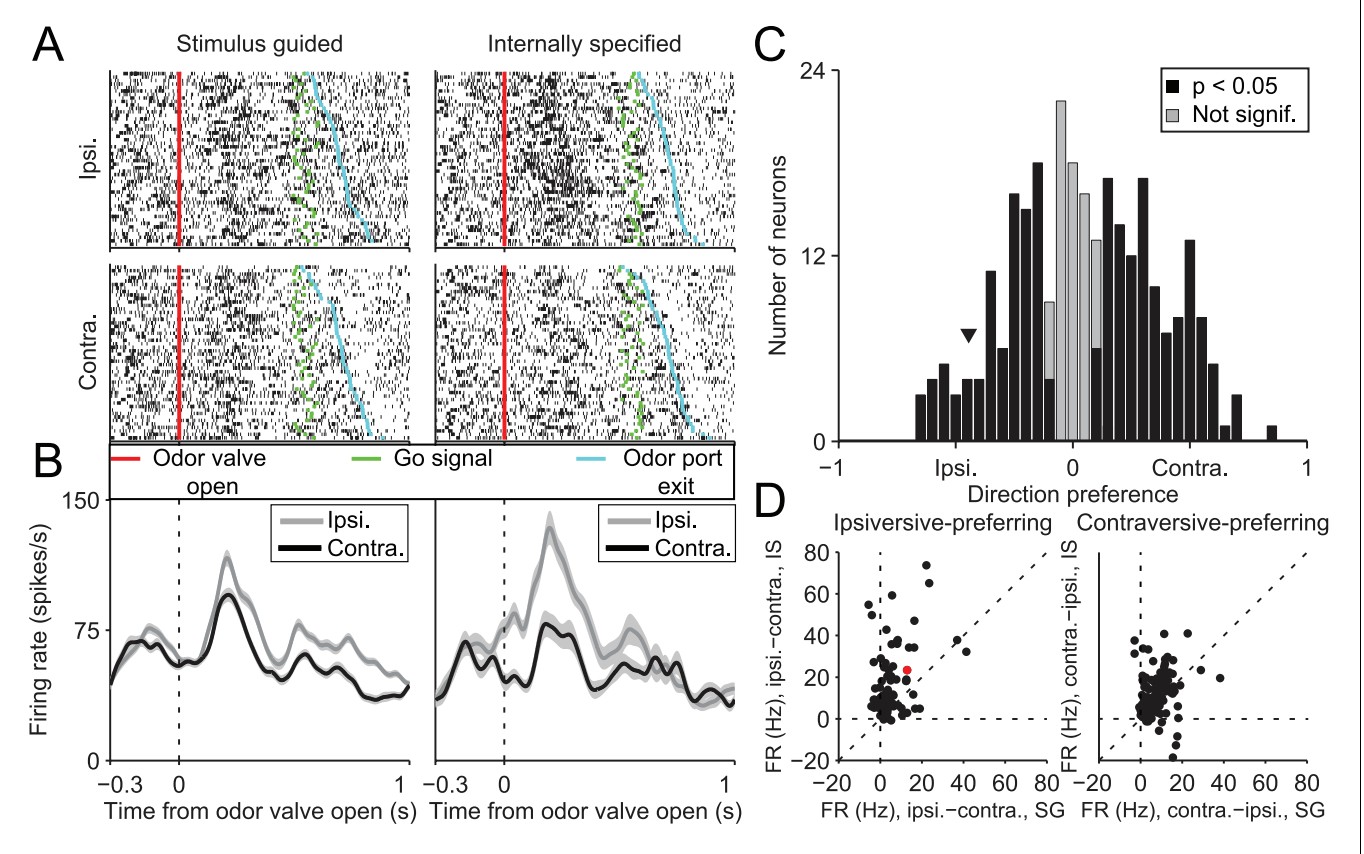

**Figure 3.** SNr activity during the delay epoch depends on movement direction and trial type. (**A**) Rasters for an example neuron grouped by movement direction (rows) and trial type (columns). For each raster, each row shows spikes (black ticks) in one trial, aligned to time of odor valve open (red line) and sorted by duration of delay epoch. Green ticks, times of go signal; blue ticks, times of odor port exit. Fifty pseudo-randomly selected trials are shown per group. (**B**) Peri-event histograms showing average activity, separately by direction, in stimulus-guided (left) and internally-specified (right) trials. Shading, ± SEM. Histograms are smoothed with a Gaussian filter ($\sigma$ = 15 ms). Ipsi., ipsiversive; Contra., contraversive. (**C**) Histogram of direction preferences during delay epoch across population of neurons. Arrowhead corresponds to example neuron in **A**. (**D**) Difference in delay-epoch firing rate between ipsiversive and contraversive trials in SG vs. IS trials in the same session, separately for ipsiversive-preferring neurons (left subpanel, corresponding to left black bars in **C**) and contraversive-preferring neurons (right subpanel, corresponding to right black bars in **C**). Only correct trials are included; all choices on 50/50 SG trials were considered correct regardless of whether they were rewarded. Dashed lines show x = 0, y = 0, and x = y. Red marker corresponds to example neuron from **A** and **B**. FR, firing rate.

that neurons were about equally likely to prefer upcoming ipsiversive and contraversive movements (*Figure 3C*), their activity depended, in a predictable manner, on trial type (*Figure 3D*).

While these findings suggest that the BG differentially mediate internally-specified and stimulus-guided movements, as we had predicted, a few differences between SG and IS trials may have contributed to this observation. We therefore sought to identify these differences and determine their influence, in several ways. First, we reasoned that, if neural activity indeed reflects trial type, firing rate would systematically change during the IS block as the mouse increasingly based its movement choice on internal representations instead of the stimulus (recall that the transitions from SG to IS blocks were unsignaled). To test this idea, we calculated the correlation between the trial-by-trial firing rate during the delay epoch and the extent to which the mouse had learned that its movement choice should be internally specified, estimated with a reinforcement learning algorithm (see Materials and methods). We performed this analysis on the 158 neurons with firing rates that depended on both direction and trial type, separately for choices in the preferred and antipreferred direction. *Figure 4A* shows data from an example neuron displaying a significant correlation for trials in the preferred direction of the neuron (r = 0.65, p = $7.0 \times 10^{-9}$), and no correlation for trials in the anti-preferred direction (r = 0.066, p = 0.60). Overall, 77/158 neurons exhibited a significant correlation

**Table 1.** Direction preference and activity change during delay epoch. Neurons are grouped by direction preference and whether activity in the preferred direction increased or decreased relative to baseline (see Materials and methods), during the delay epoch. Numbers and percentages of grand total (279) are shown; note that 17 neurons exhibited no change in activity and are not included here.

| Preference | Increase | | Decrease | | Total | |
|---|---|---|---|---|---|---|
| Contraversive | 88 | 32% | 27 | 10% | 115 | 41% |
| Ipsiversive | 52 | 19% | 38 | 14% | 90 | 32% |
| Nonselective | 48 | 17% | 26 | 9% | 74 | 27% |
| Total | 188 | 67% | 91 | 33% | 279 | 100% |

(p<0.05) between firing rate and the number of consecutive correct trials for either direction [*Figure 4B,C*; 35/77 for trials in the preferred direction (red circles), 29/77 for trials in the antipreferred direction (blue circles), and 13/77 for trials in both directions (purple circles)], with more positive correlations for trials in the preferred direction (p = $2.4 \times 10^{-6}$, $X^2$ test) and negative correlations for trials in the antipreferred direction (p = $5.9 \times 10^{-6}$, $X^2$ test), as we would expect given the pattern of results shown in *Figure 3D*. These results support the idea that SNr activity reflects the degree to which movements are selected based on internal representations.

## Modulation of SNr activity by task-relevant variables

We next examined the potential influence of other factors on the observed difference in neural activity during SG and IS trials. One difference between these trial types, by design, is that in IS trials the decision (to move left or right) is relatively easy, while in some SG trials this decision is more difficult (*Figure 1D*). The difficulty of this decision – or an associated variable, such as uncertainty, or the estimated value of each movement direction – could, in principle, affect SNr activity. Were this the case, we would expect to observe a difference in activity between those SG trials requiring an easy discrimination (mixture ratios of 95/5, 80/20, 20/80, and 5/95) and those SG trials requiring a 'difficult' discrimination (mixture ratios of 60/40, 50/50, and 40/60), since easy trials resulted in a larger fraction of correct choices (p = $3.4 \times 10^{-27}$, paired t-test; see *Figure 1D*), corresponding to a higher likelihood of reward. We therefore compared firing rate during the delay epoch between easy and difficult SG trials, separately for trials in the ipsiversive (*Figure 5A*) and contraversive (*Figure 5B*) direction, for the 216 direction-selective neurons (*Figure 3C*, black bars). We found that the activity of some individual neurons depended on difficulty (or an associated variable) (ipsiversive direction: 39/216 neurons; contraversive direction: 31/216 neurons, p<0.05, 1-way ANOVA across mixture ratios, *Figure 5A,B*), as would be predicted by the value-biasing view of BG function. However, there was little overlap (purple circles) between this small population of difficulty-dependent neurons (blue circles) and those neurons that we classified as trial-type-dependent [ipsiversive direction: 21/109 trial-type-dependent, and 18/107 non-trial-type-dependent, neurons exhibited difficulty dependence (these ratios did not differ: p = 0.32, $X^2$ test); contraversive direction: 16/101 trial-type-dependent, and 15/115 non-trial-type-dependent, neurons exhibited difficulty dependence (these ratios did not differ: p = 0.56, $X^2$ test); p<0.05, 1-way ANOVA across mixture ratios]. These results suggest that differences in decision difficulty, uncertainty, and the value associated with the direction of movement do not account for trial-type dependence or the differences in activity between SG and IS trials shown in *Figure 3D*.

We then examined whether reaction time (which differs between trial types; *Figure 1F*) and the choice on the previous trial (which, by design, is more likely to correlate with the current choice in IS blocks than SG blocks; *Figure 1C*) could explain the difference in activity between SG and IS trials. Preliminary analyses of each of these factors in isolation indicated that, as opposed to discrimination difficulty or an associated variable such as value (*Figure 5*), they often correlated with firing rate during the delay epoch. In order to determine the relative influence of these factors, as well as other factors that correlate with firing rate – current choice (*Figure 3C*) and trial type (*Figure 3D*) – on neural activity during the delay epoch, we performed a linear regression analysis with previous choice, current choice, trial type and reaction time as predictor variables (see Materials and methods). By

considering all of these factors simultaneously, this analysis provides an unbiased method for determining their influence on neural activity.

Across our population of neurons, the vast majority were influenced by at least one of these factors (281/296, p<0.05), and we found neurons with firing rates influenced by all possible combinations of factors (*Figure 6A*). Consistent with the results shown in *Figure 3C and D*, respectively, this analysis confirms that, as the mouse is selecting its direction of movement, the activity of many SNr neurons was modulated by current choice (167/296) and trial type (142/296). We also found that the activity of many neurons depended on reaction time (111/296). Surprisingly, the largest fraction of neurons exhibited activity modulated by previous choice (188/296). This is particularly interesting because this variable is critical for determining, in an IS block, which direction is associated with reward.

Given that movements can initially be selected earlier in IS than SG trials (*Figure 3A,B*), we wondered how firing rate at other times during the trial depended on previous choice, current choice, trial type, and reaction time. We therefore extended our regression analysis to examine how firing rate is modulated by these factors during overlapping 100 ms windows throughout the trial (see Materials and methods). In the example shown in *Figure 6B*, the activity of the neuron is modulated by previous choice (cyan line) – i.e., the confidence interval (shading) for this coefficient does not include 0 – even before the odor is delivered (odor valve open), and this influence persists until the movement is initiated (odor port exit). The current choice (black line) does not influence neural activity until ramping up just prior to movement initiation, but then continues to exert an influence for the remainder of the trial. The trial type (magenta line), meanwhile, exerts a moderate influence on the firing rate – specifically, activity is higher for IS trials – until just before movement initiation, after which this influence is diminished. Reaction time was a relatively poor predictor of firing rate (not shown, for clarity). To examine the dynamics of the weights of these factors across our population of neurons, we calculated the fraction of neurons with significant weights in each time window (*Figure 6C*). The pattern of results was similar to that shown in the example neuron (*Figure 6B*). Before the odor is delivered, the firing rates of about half of the neurons are influenced by the previous choice. However, as the odor is sampled, the influence of the previous choice decreases and the influence of the current choice increases, with about two thirds of neurons exhibiting a significant weight for the current choice by the time the movement is initiated. Interestingly, trial type modulates the activity of about one third of neurons throughout the trial. The influence of reaction time is

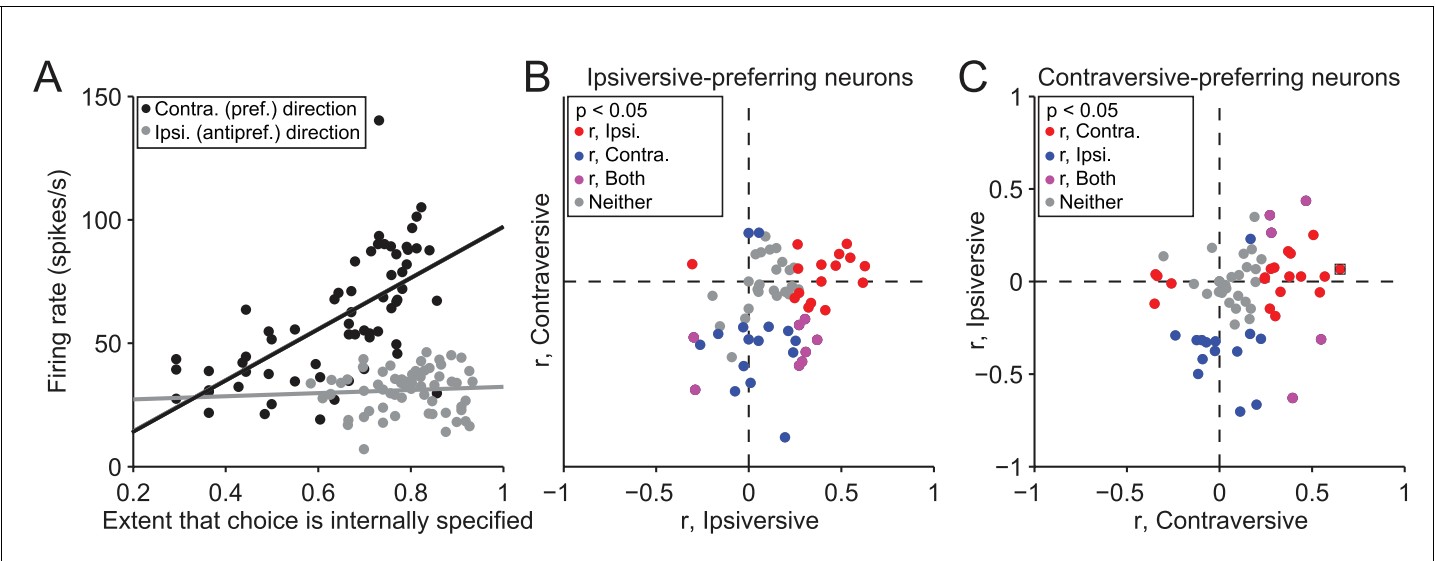

**Figure 4.** Activity depends on the extent to which movements are internally specified. (**A**) Firing rate during delay epoch plotted as a function of the value of the rewarded side, estimated via reinforcement learning ($V_{dir, t}$), for both IS blocks in a session, for one example neuron. Each circle corresponds to one trial. (**B**) Correlations (as in panel **A**) for ipsiversive movement plotted against contraversive movement, for the population of ipsiversive-preferring neurons (left black bars in *Figure 3C*) with activity that depended on trial type (SG vs. IS). Each circle corresponds to one neuron. (**C**) Same as **B**, for contraversive-preferring neurons (right black bars in *Figure 3C*). Black box corresponds to example neuron from **A**.

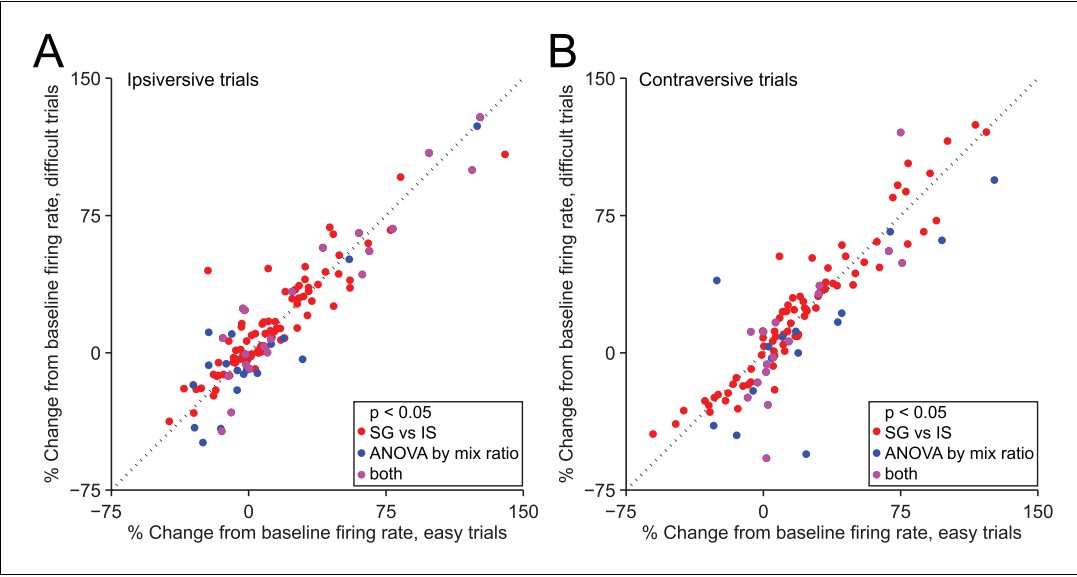

**Figure 5.** Dependence of firing rate on trial type cannot be explained by discrimination difficulty or an associated variable. (**A**) Mean normalized change from baseline (Fc, see Materials and methods) during delay epoch of easy vs. difficult ipsiversive SG trials of direction-selective neurons (black bars in *Figure 3C*). Each circle corresponds to one neuron. Red circles indicate that activity differs between SG and IS trials, and does not depends on mixture ratio (or an associated variable such as discrimination difficulty). (**B**) Same as **A**, for contraversive trials.

strongest during movement but has relatively little influence on the population (not shown). These results indicate that SNr activity dynamically reflects trial type and other task-relevant variables throughout the trial, as would be expected if the BG are differentially involved in mediating stimulus-guided and internally-specified movements.

## Discussion

We have shown that SNr activity preceding orienting movements depends on whether the direction of movement was indicated by a stimulus or was specified by internal variables (*Figures 3*, *4*). While we designed the task such that correct movements were equally valuable across these conditions (*Figure 1*), given imperfect (and stochastic) choice behavior, the experienced value was not necessarily identical. However, the dependence on trial type could not be accounted for by differences in the estimated value of each movement direction – or an associated variable, such as difficulty or uncertainty in selecting the movement – between the trial types (*Figure 5*). In some neurons this dependence could be explained, in part, by the choice on the previous trial (*Figure 6A*), which is informative of the rewarded direction in IS blocks. Over the course of the trial, while the influence on SNr activity of the previous choice decreased and that of the current choice increased, as might be expected given the demands of the task, the influence of trial type remained relatively constant (*Figure 6C*). These results suggest that the SNr is differentially engaged by stimulus-guided and internally-specified movements.

Previous studies in primates have shown that movement-related SNr activity was higher for memory-guided than visually-guided saccades (*Hikosaka and Wurtz, 1983a*), and that SNr stimulation had a larger effect on memory- than visually-guided saccades (*Basso and Liu, 2007*). Movements selected based on a remembered stimulus can be thought of as internally specified, and in this sense our results (*Figure 3D*) are consistent with these findings and demonstrate that they generalize across species and movement types (saccades and full-body orienting). However, the rewarded direction in both memory- and visually-guided trials was indicated by the stimulus, which was not the case in our IS trials, in which we sometimes observed that direction preference emerged even before stimulus delivery (see example in *Figure 3A,B*). Further, the difference between direction preference in IS and SG trials during the delay epoch was correlated with the difference between preference in

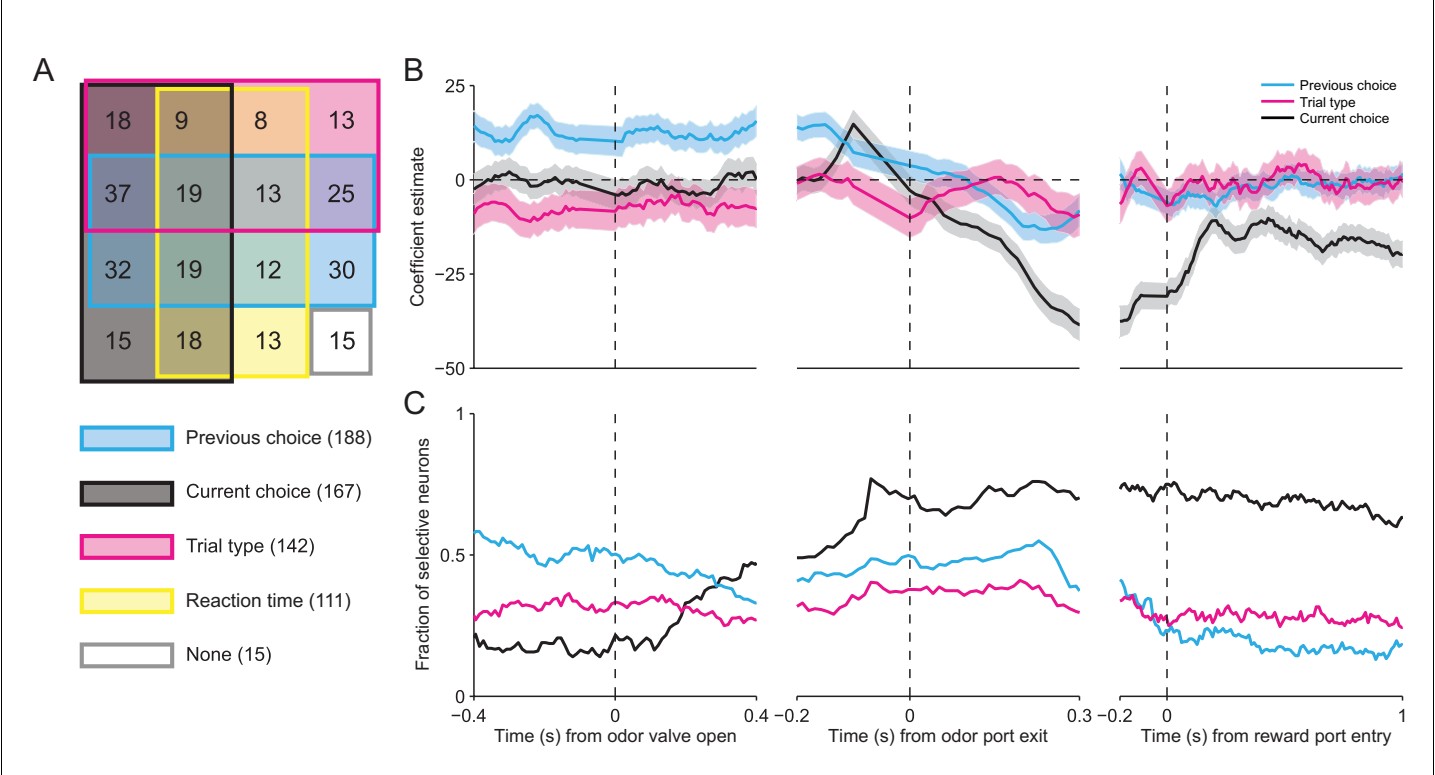

**Figure 6.** SNr activity is influenced by several task-relevant factors throughout the trial. (**A**) Venn diagram showing the number of neurons whose firing rate during the delay epoch was significantly influenced (p < 0.05) by previous choice, current choice, trial type, reaction time, and all combinations of these factors, or by no factor. (**B**) $\beta$ coefficients estimated based on firing rate in 100 ms bins aligned to three different trial events for one example neuron (reaction time coefficient not shown, for clarity). Shading, ± 95% confidence interval. (**C**) Fraction of neurons with a significant $\beta$ coefficient corresponding to each predictor variable in each 100 ms bin, aligned as in panel **B**. All 296 neurons were included in this analysis.

IS and SG trials during the epoch from odor port entry to odor valve open (r = 0.47, p = 4.8 × $10^{-18}$). These results demonstrate that, in IS trials, the direction of movement was initially selected independent of the stimulus, which contributes to the difference in activity during the delay epoch that we observe between SG and IS trials. In addition, while *Hikosaka and Wurtz (1983a)* examined only neurons that exhibited a decrease in activity around the time of contraversive saccades, we examined increasing and decreasing neurons that prefer both ipsiversive and contraversive movement (*Gulley et al., 2002*, *1999*; *Handel and Glimcher, 2000*; *Sato and Hikosaka, 2002*) and found that all of these groups exhibited a difference in activity between stimulus-guided and internally-specified movements. Therefore, the differences we observed between internally-specified and stimulus-guided movements extend our understanding of SNr function.

Interestingly, patients with Parkinson's disease and other BG pathologies have been reported to exhibit greater deficits in the initiation of internally-specified than visually-guided movements (*Forssberg et al., 1984*; *Laplane et al., 1984*; *Azulay et al., 1999*). While the neural basis for this phenomenon is not well understood and remains an active area of study (*Distler et al., 2016*), it has been suggested that visual cues engage (intact) sensorimotor pathways outside of the BG, such as the cerebellum (*Glickstein and Stein, 1991*). Our results suggest that differential processing of internally-specified and visually-guided movements within the BG themselves may also contribute to this clinical observation.

As noted above, other studies have found that movement-related SNr activity is modulated by the relative value associated with a movement (*Bryden et al., 2011*; *Sato and Hikosaka, 2002*), including whether the movement will be rewarded at all (*Handel and Glimcher, 2000*). This value dependence likely arises from dopaminergic input to the BG that is thought to convey reward-related information (*Schultz et al., 1997*), and has been accounted for by a model in which, prior to

stimulus presentation, reward expectation modulates striatal inputs to the SNr in order to bias downstream superior colliculus (SC) activity such that the most valuable movement is facilitated (*Hikosaka et al., 2006*; *Wolf et al., 2015*). We propose that a similar model can also explain how internally-specified movements, more generally, are facilitated (*Figure 7*).

To illustrate this idea, consider how, given the data presented here, the relative activity between ipsiversive-preferring left and right SNr neurons would relate to an upcoming rightward movement (we consider relative, rather than absolute, activity since this is most directly relevant to the decision – move left vs. move right – required by our task). Left and right SNr neurons would exhibit a larger difference in activity in IS trials than in SG trials (*Figure 3D*, left). If ipsiversive-preferring SNr neurons primarily project to the ipsilateral SC (*Hikosaka and Wurtz, 1983b*), then a downstream left SC neuron, the activity of which promotes rightward movement (*Felsen and Mainen, 2012*; *Horwitz and Newsome, 2001*; *Stubblefield et al., 2013*) will receive less inhibition from the left SNr when the movement is internally specified, thereby facilitating rightward movements that are internally specified (*Figure 7*). Preceding the same movement, contraversive-preferring left and right SNr neurons would also exhibit a larger difference in activity in IS trials than in SG trials (*Figure 3D*, right). If contraversive-preferring SNr neurons comprise the 'crossed' projection to the contralateral SC (*Jiang et al., 2003*), then a downstream right SC neuron would receive more inhibition from the left SNr when the movement is internally specified, again facilitating the rightward movement. However, we observed that slightly more SNr neurons prefer contraversive than ipsiversive movement (*Figure 3C*) but many fewer SNr neurons project to the contralateral than ipsilateral SC, particularly in rodents (*Beckstead et al., 1981*; *Deniau et al., 1977*; *Gerfen et al., 1982*; *Jayaraman et al.,*

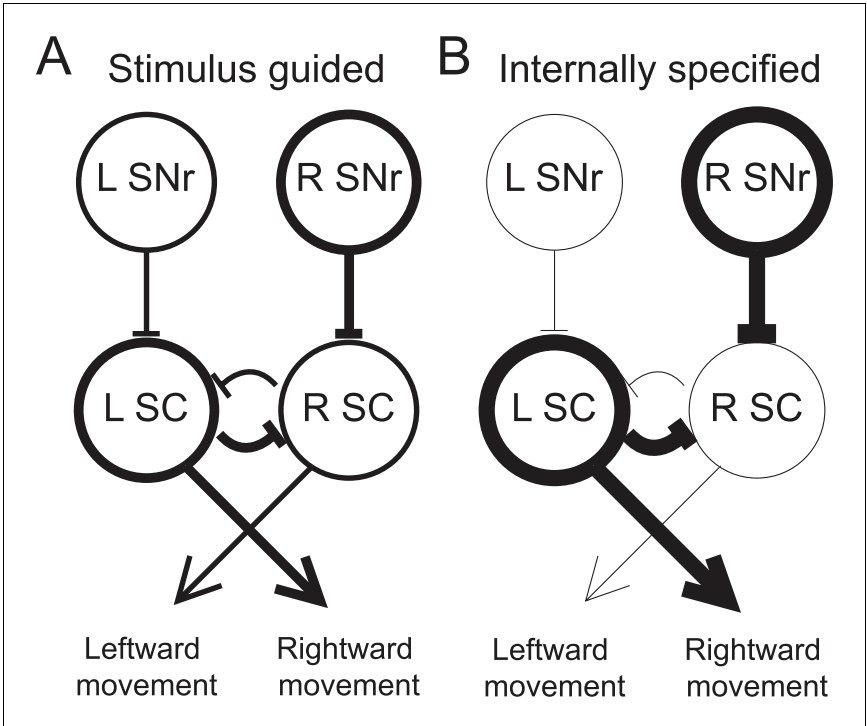

**Figure 7.** Model proposing how the observed activity of ipsiversive-preferring SNr neurons could facilitate internally-specified movements relative to stimulus-guided movements. (**A**) Line thickness corresponds to level of activity. Activity preceding stimulus-guided rightward movement. A left SNr neuron is moderately weakly active, providing moderately weak inhibition to the left SC (superior colliculus). A right SNr neuron is moderately strongly active, providing moderately strong inhibition to the right SC. This pattern of activity in the SC moderately promotes rightward movement. (**B**) Activity preceding internally-specified rightward movement. Compared to **A**, a left SNr neuron is very weakly active, providing very weak inhibition to the left SC; and a right SNr neuron is very strongly active, providing very strong inhibition to the right SC. This pattern of activity in the SC strongly promotes rightward movement.

*1977*), and contraversive-preferring SNr neurons may preferentially project to non-tectal targets. Thus, SNr activity may be consistent with the facilitation of internally-specified contraversive movements. Our results therefore extend the model underlying the value-biasing view of BG function (*Hikosaka et al., 2006*) by suggesting that the influence of the SNr on downstream motor regions is modulated by internal representations in addition to value.

In summary, we have shown that SNr activity depends on whether otherwise-identical movements are specified by internal representations of task variables or guided by an external stimulus. We suggest that this dependence may reflect a facilitation for internally-specified movements, consistent with the view that, although movements are often made in response to sensory stimuli, internal representations of priors play a critical role in guiding motor output (*Wolpert and Landy, 2012*). Our results are sufficiently consistent with results in primate SNr (*Handel and Glimcher, 2000*, *1999*; *Hikosaka and Wurtz, 1983a*; *Liu and Basso, 2008*; *Sato and Hikosaka, 2002*) that they can inform the interpretation of previous studies (e.g., our proposed extensions of the model explaining the value-biasing role of the BG described above), while also offering novel insight into BG function. Future studies can utilize the task established here, in the experimentally-advantageous awake-behaving mouse model (*Carandini and Churchland, 2013*), to examine whether the difference in SNr activity preceding internally-specified and stimulus-guided movements is established by local processing or via striatal inputs (*Hikosaka et al., 2006*; *Lauwereyns et al., 2002*) and to further elucidate how the BG control goal-directed movements.

## Materials and methods

### Animal subjects

All experiments were performed according to protocols approved by the University of Colorado School of Medicine Institutional Animal Care and Use Committee. We used male adult C57BL/6J mice (n = 4, determined by estimating the number of neurons required for our analyses and by the number of neurons recorded per mouse in initial experiments; aged 7–14 months at the start of experiments; Jackson Labs) housed in a vivarium with a 12-hr light/dark cycle with lights on at 5:00 am. Food (Teklad Global Rodent Diet No. 2918; Harlan) was available ad libitum. Access to water was restricted to the behavioral session to motivate performance; however, if mice did not obtain ~1 ml of water during the behavioral session, additional water was provided for ~2–5 min following the behavioral session (*Smear et al., 2011*; *Thompson and Felsen, 2013*). All mice were weighed daily and received sufficient water during behavioral sessions to maintain >85% of pre-water restriction weight.

### Behavioral task

In general, mice were first trained to perform an odor-guided spatial choice task – which was comprised of 'stimulus-guided' (SG) trials – as described in *Stubblefield et al. (2013)*, and were then trained to perform 'internally-specified' (IS) trials. Briefly, each mouse was water-restricted and trained to interact with three ports (center: odor port; sides: reward ports) along one wall of a behavioral chamber (Island Motion). In each trial, the mouse entered the odor port, triggering the delivery of an odor; waited 488 ± 104 ms (mean ± SD) for a go signal (auditory tone); exited the odor port; and entered one of the reward ports (*Figure 1A*). Premature exit from the odor port resulted in the unavailability of reward on that trial. Odors were comprised of binary mixtures of (+)-carvone and (-)-carvone, commonly perceived as caraway and spearmint, respectively; an enantiomeric odor pair was selected to control for differences in molecular structure of odorant stimuli. In each SG trial, one of seven odor mixtures was presented via an olfactometer (Island Motion): volume (+)-carvone/(-)-carvone = 95/5, 80/20, 60/40, 50/50, 40/60, 20/80, or 5/95. Mixtures in which (+)-carvone > (-)-carvone indicated reward availability only at the right port and mixtures in which (-)-carvone > (+)-carvone indicated reward availability only at the left port [we therefore refer to (-)-carvone as the 'left odor' (e.g., *Figure 1D*) for simplicity]. In trials in which (+)-carvone = (-)-carvone, the probability of reward at the left and right ports, independently, was 0.5. Reward, consisting of 4 µl of water, was delivered by transiently opening a calibrated water valve 10–100 ms after reward port entry. Odor and water delivery were controlled, and port entries and exits were recorded, using custom software (available at https://github.com/felsenlab; adapted from C. D. Brody) written in MATLAB (MathWorks).

Mice learned to perform SG trials within ~48 sessions (1 session/day); detailed training stages are described in *Stubblefield et al. (2013)*. Mice required an additional ~5 sessions to learn to perform interleaved blocks of SG and IS trials. In every IS trial the 50/50 mixture was presented, and reward was available only at one side throughout the block. Mice were first introduced to interleaved blocks, each of which required 25 correct trials to advance to the next block. Once they performed ~70% of trials in the session correctly, the number of correct trials required per block was increased to 50. Mice performed 5 blocks (SG, IS, SG, IS, SG) per session (*Figure 1B*); the side associated with reward switched between each IS block. Upon completing training, mice were implanted with microdrives for neural recording (see below). During each of the 54 recording sessions, mice performed 321.81 ± 89.49 (mean ± SD) trials.

## Surgery

Details of the surgical procedure are provided in *Thompson and Felsen (2013)*. Briefly, once the mouse was fully trained on the task, it was anesthetized with isoflurane and secured in a stereotaxic device, the scalp was incised and retracted, 2 small screws were attached to the skull, and a craniotomy targeting the left SNr was performed, centered at 3.27 mm posterior from bregma and 1.4 mm lateral from the midline (*Paxinos and Franklin, 2004*). A VersaDrive 4 microdrive (Neuralynx), containing 4 independently adjustable tetrodes, was affixed to the skull via the screws, luting (3M), and dental acrylic (A-M Systems). A second small craniotomy was performed in order to place the ground wire in direct contact with the brain. After the acrylic hardened, a topical triple antibiotic ointment (Major) mixed with 2% lidocaine hydrochloride jelly (Akorn) was applied to the scalp, the mouse was removed from the stereotaxic device, the isoflurane was turned off, and oxygen alone was delivered to the animal to gradually alleviate anesthetic state. Mice were administered sterile isotonic saline (0.9%) for rehydration and an analgesic (Ketofen; 5 mg/kg) for pain management. Analgesic and topical antibiotic administration was repeated daily for up to 5 days, and animals were closely monitored for any signs of distress.

## Electrophysiology

Neural recordings were collected using four tetrodes, wherein each tetrode consisted of four polyimide-coated nichrome wires (Sandvik; single-wire diameter 12.5 µm) gold plated to 0.2–0.4 MΩ impedance. Electrical signals were amplified and recorded using the Digital Lynx S multichannel acquisition system (Neuralynx) in conjunction with Cheetah data acquisition software (Neuralynx).

Tetrode depths were adjusted approximately 23 hr before each recording session in order to sample an independent population of neurons across sessions. To estimate tetrode depths during each session we calculated distance traveled with respect to rotation fraction of the screw that was affixed to the shuttle holding the tetrode. One full rotation moved the tetrode ~250 µm and tetrodes were moved ~62.5 µm between sessions. The final tetrode location was confirmed through histological assessment using electrolytic lesions and tetrode tracks (see below).

Offline spike sorting and cluster quality analysis was performed using MClust software (MClust-3.5, A.D. Redish, et al.) in MATLAB. Briefly, for each tetrode, single units were isolated by manual cluster identification based on spike features derived from sampled waveforms (*Figure 2B*). Identification of single units through examination of spikes in high-dimensional feature space allowed us to refine the delimitation of identified clusters by examining all possible two-dimensional combinations of selected spike features. We used standard spike features for single unit extraction: peak amplitude, energy (square root of the sum of squares of each point in the waveform, divided by the number of samples in the waveform), and the first principal component normalized by energy. Spike features were derived separately for individual leads. To assess the quality of identified clusters we calculated two standard quantitative metrics: L-ratio and isolation distance (*Schmitzer-Torbert et al., 2005*). Clusters with an L-ratio of less than 0.70 and isolation distance greater than 6.5 were deemed single units, which resulted in the exclusion of 12% of the identified clusters. Although units were clustered without knowledge of interspike interval, only clusters with few interspike intervals less than 1 ms were considered for further examination. Furthermore, we excluded the possibility of including data from the same neuron twice by ensuring that both the waveforms and response properties sufficiently changed across sessions. If they did not, we conservatively assumed that we were recording from the same neuron, and only included data from one session.

## Lesioning and histology

To verify final tetrode location we performed electrolytic lesions (100 µA, ~1.5 min per lead) after the last recording session. One day following lesion, mice were overdosed with an intraperitoneal injection of sodium pentobarbital (100 mg/kg) and transcardially perfused with saline followed by ice-cold 4% paraformaldehyde (PFA) in 0.1 M phosphate buffer (PB). After perfusion, brains were submerged in 4% PFA in 0.1 M PB for 24 hr for post-fixation and then cryoprotected for 24 hr by immersion in 30% sucrose in 0.1 M PB. The brain was encased in the same sucrose solution, and frozen rapidly on dry ice. Serial coronal sections (60 µm) were cut on a sliding microtome for reconstruction of the lesion site and tetrode tracks. Fluorescent Nissl (NeuroTrace, Invitrogen) was used to identify cytoarchitectural features of the SNr and verify tetrode tracks and lesion damage within or below the SNr. Images of SNr (see *Figure 2A*) were captured with a 10x objective lens, using an LSM 5 Pascal series Axioskop 2 FS MOT confocal microscope (Zeiss).

## Analyses and statistics

All analyses were performed in MATLAB.

### Direction preference

To quantify the selectivity of single neurons for movement direction, we used an ROC-based analysis (*Green and Swets, 1966*). This analysis calculates the ability of an ideal observer to classify whether a given spike rate was recorded in one of two conditions (here, preceding leftward or rightward movement). We defined 'preference' as $2(ROC_{area} - 0.5)$, a measure ranging from $-1$ to 1, where $-1$ signifies the strongest possible preference for left, 1 signifies the strongest possible preference for right, and 0 signifies no preference (*Feierstein et al., 2006*). Statistical significance was determined with a permutation test: we recalculated the preference after randomly reassigning all firing rates to either of the two groups arbitrarily, repeating this procedure 500 times to obtain a distribution of values, and calculated the fraction of random values exceeding the actual value. We tested for significance at $\alpha$ = 0.05. Trials in which the movement time (between odor port exit and reward port entry) was > 1.5 s were excluded from all analyses. Neurons with fewer than 100 trials of each type (SG and IS) or with a firing rate below 2.5 spikes/s for either trial type or across the entire session (Fc, described below), were excluded from all analyses.

### Sign of activity change during delay epoch

We calculated the normalized response (NR) for each neuron as NR = Ft/Fc where Ft is the mean firing rate in the 'test' window (delay epoch) and Fc is the mean firing rate in the 'control' window (*Sato and Hikosaka, 2002*) across all trials in the preferred direction of the neuron (or, for neurons with no direction preference, across all trials). Since the structure of our task does not include a natural 'control' epoch – i.e., in which the animal is in a motionless state unaffected by task demands – our control window was defined as the time of odor port entry to reward port exit (i.e., the duration of the trial). Neurons with NR < 1 were defined as decreasing and neurons with NR > 1 were defined as increasing (*Table 1*). Note that, by convention, a decreasing neuron that decreases more for contraversive than ipsiversive movement would be considered to have an ipsiversive direction preference (as calculated above), because firing rate is higher for ipsiversive movement (*cf. Sato and Hikosaka, 2002*).

### Reinforcement learning model

In order to estimate the value associated with each direction of movement, we iteratively updated the value of each direction in each trial as $V_{dir,t} = V_{dir,t-1} + \alpha(R_{dir,t-1} - V_{dir,t-1})$, where $R_{dir,t-1}$ is the reward for the given direction in the previous trial in which that direction was chosen [0 for unrewarded (which includes trials in which the correct choice was made but the odor port was exited before the go signal) and 1 for rewarded] and $\alpha$ is the learning rate (we set $\alpha = 0.1$; values of 0.03 and 0.3 did not affect the results). The value of each direction was updated independently. This estimate is based on the Q-learning algorithm (*Sutton and Barto, 1998*; note that we excluded a term for maximum future value because this was independent of the choice on the current trial). $V_{dir,t}$ therefore ranged from 0 to 1. Since we calculated $V_{dir,t}$ in each trial of the session (including SG and IS trials), it tended to start near 0.5 (but was not exactly 0.5) at the beginning of each IS block. In

well-behaved IS blocks, $V_{dir,t}$ tended to asymptotically approach 1 as the mouse consistently returned to the rewarded port. $V_{dir,t}$ was calculated separately for the ipsiversive and contraversive directions and was only updated in trials in which that direction was selected.

### Regression model

To assess the influence of several factors on SNr activity (*Figure 6*), we fit the electrophysiological data with a multi-variable linear regression model of the form

$$FR = \beta_0 + \beta_{Previous\ choice}\chi_{Previous\ choice} + \beta_{Current\ choice}\chi_{Current\ choice} + \beta_{Trial\ type}\chi_{Trial\ type} + \beta_{Reaction\ time}\chi_{Reaction\ time},$$

where $FR$ is the firing rate during the delay epoch,

$$\chi_{Previous\ choice} = \begin{cases} -1 & \text{for an ipsiversive choice} \\ 0 & \text{for no choice} \\ 1 & \text{for a contraversive choice} \end{cases} \text{in the previous trial,}$$

$$\chi_{Current\ choice} = \begin{cases} -1 & \text{for an ipsiversive choice} \\ 0 & \text{for no choice} \\ 1 & \text{for a contraversive choice} \end{cases} \text{in the current trial,}$$

$$\chi_{Trial\ type} = \begin{cases} -1 & \text{for an IS trial} \\ 1 & \text{for an SG trial} \end{cases},$$

$$\chi_{Reaction\ time} = \text{time from go cue to reward port entry (normalized between 0 and 1),}$$

$\beta_0$ represents the mean firing rate across trials during the delay epoch, and $\beta_{Previous\ choice}$, $\beta_{Current\ choice}$, $\beta_{Trial\ type}$, and $\beta_{Reaction\ time}$ represent the influence on firing rate of the previous choice, the current choice, the trial type, and the reaction time corresponding to that trial. Positive values for $\beta_{Previous\ choice}$ and $\beta_{Current\ choice}$ indicate that firing rate is increased by contraversive choices and negative values indicate that firing rate is increased by ipsiversive choices. Positive values for $\beta_{Trial\ type}$ indicate that firing rate is increased by stimulus-guided trials and negative values indicate that firing rate is increased by internally-specified trials. The sign of $\beta_{Reaction\ time}$ indicates the sign of the correlation between reaction time and firing rate. We used the MATLAB function fitlm to estimate the $\beta$s and calculate their significance and confidence intervals. We also performed this same regression analysis with two additional terms, for value [estimated in IS trials with the reinforcement learning model (see above) and in SG trials as the average performance by mixture ratio within the block (*Figure 1D*)], and the interaction between value and trial type. We found that, while the firing rate of some neurons was influenced by these additional factors, as expected given the value-biasing view of BG function (*Hikosaka et al., 2006*), including them – or including the log of the value – did not affect the overall results shown in *Figure 6A*. To examine how firing rate throughout the trial depended on these factors (*Figure 6B,C*), we repeated the above analysis with respect to firing rate in overlapping 100 ms bins, shifted by 10 ms, aligned to three behavioral events: odor valve open, odor port exit and reward port entry.

## Acknowledgements

We thank Michele Basso, Abigail Person, Jaclyn Essig, Andrew Wolf, and Elizabeth Stubblefield for comments on the manuscript; Angie Ribera for use of the confocal microscope; and John Thompson, Jamie Costabile, and Quang Dang for experimental and technical assistance. This work was supported by the National Institutes of Health (R01 NS079518).

## Additional information

### Funding

| Funder | Grant reference number | Author |
| --- | --- | --- |
| National Institute of Neurological Disorders and Stroke | R01NS079518 | Gidon Felsen |

The funders had no role in study design, data collection and interpretation, or the decision to submit the work for publication.

## Author contributions

MJL, Designed the experiments, Collected the data, Analysis and interpretation of data, Wrote the manuscript; GF, Designed the experiments, Analysis and interpretation of data, Wrote the manuscript

## Author ORCIDs

Gidon Felsen, http://orcid.org/0000-0003-0745-8279

## Ethics

Animal experimentation: This study was performed in accordance with the recommendations in the Guide for the Care and Use of Laboratory Animals of the National Institutes of Health, 8th edition. All experiments were performed according to protocol #90209(12)1D, approved by the University of Colorado School of Medicine Institutional Animal Care and Use Committee. All surgeries were performed under isoflurane anesthesia and all perfusions were performed following an overdose of sodium pentobarbital. Quality of life was improved with enriched living environments and dietary treats while every effort was made to minimize suffering.

# Additional files

## Supplementary files

• Supplementary file 1. Table of mean firing rates, p-values, and preference values for each SNr neuron included in the study. Rows correspond to individual neurons. Columns B-C show mean and standard deviation of activity during the baseline (our 'control' window, from odor port entry to reward port exit). Column D shows mean firing rates during the odor sampling epoch (i.e., delay epoch) in leftward (ipsiversive) SG trials. Column E shows p-values for paired t-tests of activity in leftward trials during the odor sampling epoch compared against baseline. Column F is similar to D, for rightward (contraversive) SG trials. Column G is similar to E, for rightward SG trials. Columns H-M are similar to B-G, for IS trials. Columns N-P show p-values for unpaired t-test comparing between SG and IS trials for baseline, odor sampling epoch in leftward trials, and odor sampling epoch in rightward trials, respectively. Column Q shows preference values calculated during the odor sampling epoch and Column R shows p-values associated with these preference values (corresponding to *Figure 3C*).

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
