## [Decision Letter]

Thank you for submitting your work entitled "Basal ganglia output reflects internally-specified movements" for consideration by *eLife*. Your article has been reviewed by two peer reviewers, and the evaluation has been overseen by a Reviewing Editor and Timothy Behrens as the Senior Editor.

The reviewers have discussed the reviews with one another and the Reviewing Editor has drafted this decision to help you prepare a revised submission.

The reviewers and reviewing editor agree that the study clearly presents an interesting and well done study, where the authors show that activity in SNr, the main basal ganglia output nucleus in rodents, signals preferentially internally-generated movements rather than stimulus-driven movements. The results are consistent with the clinical findings in basal ganglia disorders (self-paced movements are more affected than stimulus-driven ones), but are still novel in the sense that they show that SNr activity precedes many times internally generated movements.

However, there are several issues to be addressed before the manuscript can be considered further. Please find attached below the original comments and also a summary of the most important issues to take into account.

1) The reviewers suggest new analyses to quantify the activity of single SNr neurons in the different conditions. They specifically suggest that the authors analyse the activity of individual neurons during ipsi versus contralateral movements.

2) The reviewers ask for new analyses to demonstrate that activity preferentially encodes internally-generate movements versus stimulus-guided throughout different epochs of the movement. In the current version of the study, the activity is measured at constant times in the trial which, given the fact that reaction times are different, may mean different epochs in relation to movement are being compared.

3) The reviewers also comment that SNr-SC connections are largely ipsilateral in different species. The data presented almost suggests that the contralateral connection is equally common. How do the authors explain this?

4) The reviewers propose other ways of quantifying and analysing/fitting the data; namely using a simple RL model.

5) The flow of the manuscript tries to set up an opposition between value-based action selection or movement biasing, versus internally generated action selection or movement biasing. However, the study does not address that, it rather addresses the comparison between similarly valued stimulus-driven versus internally-guided movements (in other formulations feedforward versus feedback driven, sensorimotor versus motor sensory). Therefore, the study does not address that expected value does not influence the activity of SNr during the execution of similar internally-guided movements. It does compare the activity of similarly valued movements, internally-guided are more represented than stimulus-driven. The study is not contrary to the value-biasing view of basal ganglia function, it just shows that basal ganglia cares more about self-paced that stimulus-driven movements (even for the same value). Therefore, the Abstract; Introduction, Discussion, and flow should focus on the difference between stimulus-driven versus internally-guided movements.

Specific Comments:

Reviewer #1:

6) This study provides data suggesting that the basal ganglia contribute to the internally triggered motor behavior. It has been known that patients with basal ganglia dysfunction may have difficulty in initiating movements internally. For example, patients with Parkinson's disease often have difficulty in initiating movements (e.g., walking) if there is no sensory guidance (Glickstein & Stein, Trends Neurosci 1991) and patients with bilateral pallidal lesions may not initiate goal-directed behavior spontaneously (Laplane et al., J Neurol Neurosurg Psychiatry 1984). However, the underlying mechanism is still unclear. In this sense, this study is important.

A difficulty of this theme is how to guide the subject to initiate movements internally. Without any sensory guidance, motor behavior is likely uncontrollable and, if so, behavioral as well as neuronal data may not be feasible for statistical analysis. In such a difficult context, the authors devised a simple but controllable procedure for mice. Thus, this manuscript is valuable.

With my high expectation, I now have many suggestions and comments, as shown below.

It is important to discuss the results of this study in relation to the well-known clinical symptoms described above.

7) The authors analyzed the preferred and antipreferred directions separately throughout the description of the results, which makes the interpretation very complex and difficult. In the end, to interpret the data as a whole the authors focused on the difference in neuronal activity between the contralateral and ipsilateral choices (Figure 7). Aiming at this goal, it would be much better to use this measure (i.e., Ipsi-Contra) from the beginning. Below are my suggestions.

8) Figure 3: Rearrange graphs in a 2x2 format: SG (left) vs. IS (right), Ipsi (top) vs. Contra (bottom). The Ipsi-Contra difference would become more evident in IS, but not SG.

9) Figure 3: In the current version, it is unclear how individual neurons behaved. My suggestion is to plot the Ipsi-Contra difference for each neuron. The distribution of data points would be similar to the current graph. I would then analyze data statistically, including populational trend (e.g., Ipsi-Contra difference is larger in IS than SG) and classification of individual neurons (e.g., IS-preferring, SG-preferring).

10) Figure 4: In the current version, 4A has the same issue as for Figure 1, and 4B is almost meaningless. I would use the Ipsi-Contra difference for 4A for each neuron. Then, show the across-trial changes (perhaps, using a regression line) for a number of neurons; in this case, it may be useful to use the absolute value of the Ipsi-Contra difference, because it would increase across trials. My other suggestion is to plot the Ipsi-Contra difference across time within a trial; the value would increase for IS than SG (as in Figure 3).

11) Figure 5: The current version includes multiple comparisons, although the graphs do not visually show the results of the comparisons. My suggestion is to present simple histograms showing the absolute values of the Ipsi-Contra difference, separately for IS, SG (easy), and SG (difficult).

12) Figure 6: This figure is too complicated and does not add any support for the conclusion. Moreover, the data include both SG and IS, which I find no benefit. The previous choice should have a larger effect in IS than SG. The current choice should start earlier for IS than SG. My suggestion is to do the same analysis separately for SG and IS, although it may provoke more issues.

13) I have to add one issue to my own suggestion – focus on the Ipsi-Contra difference. The feature that is orthogonal to this measure is the average magnitude of the Ipsi & Contra activity. If this is high, SC on both sides would be inhibited and the action (e.g., orienting) would be suppressed in general. If low, the action would be enhanced in general. The authors need to consider if this feature contributes to any difference in animals' choice behavior in general.

Below, I add my comments (rather than suggestions), which can be critical.

14) Activity of SNr neurons started differential (ipsi vs. contra) earlier in IS trials (even before odor delivery) than SG trials (basically after odor delivery), as exemplified in Figure 3. Now, the magnitude of neuronal activity was measured during the delay epoch that started with odor delivery. This raised the possibility that the differential activity was higher in IS trials than SG trials simply because the differential activity started earlier in IS trials. Please answer this question.

15) The different connections shown in Figure 7 may explain the functions of ipsiversive and contraversive SNr neurons, but this is a hopeful scheme with no evidence or hints. Virtually all anatomical studies have shown that the majority of the SNr-SC connections are ipsilateral in various mammals. The contralateral SNr-SC connection is very few in rats (Beckstead et al., J Neurosci 1981), which may be true in mice. The authors found that contraversive SNr neurons (thus, projecting to the contralateral SC) are slightly more common than ipsiversive SNr neurons (thus, projecting to the ipsilateral SC). This raises a concern about the scheme in Figure 7.

Other comments and questions:

16) Results: "if value is only one of a number of internal representations contributing to movement selection, then these movements would differentially engage the BG."

What does this mean?

17) Results: "We recorded from 298 well isolated left SNr neurons"

Why did you record SNr neurons only on the left side in 4 mice?

18) Results: "while we found that neurons were equally likely to prefer upcoming ipsiversive and contraversive movements (Figure 3), they were more likely to exhibit a higher firing rate preceding IS movements in their preferred direction and preceding SG movements in their antipreferred direction (Figure 3)."

It is unclear what this sentence means.

19) Results: "We observed no difference in activity between SG trials in which different mixtures were presented but movement direction was held constant"

Please show statistical evidence.

20) Results: "The difficulty of this decision could, in principle, affect SNr activity." Another possibility is 'uncertainty', rather than difficulty.

21) Results: "there was little overlap between this small population of difficulty-dependent neurons and those neurons that exhibited a dependence on trial type (Figure 5)". This notion is unclear simply by looking at these graphs.

Reviewer #2:

The authors present results from a study in which they recorded neural activity in the SNr while mice carried out a task in which they chose left or right reward ports either on the basis of an odor mixture (stimulus guided) or on the basis of previously rewarded outcomes (internally-guided). They examined neural activity, mostly during the choice period and found that SNr neurons coded ipsiversive and contraversive movements with about equal frequency. Neurons responded more strongly for internally-guided movements in their preferred direction, and more strongly for stimulus guided movements in their anti-preferred direction. They also correlated with learning in the IS blocks.

This study addresses an interesting question about the role of the basal ganglia, assessed from the perspective of an output nucleus, in internally generated vs. externally cued movements. Overall the study was well carried out. The dataset is reasonable and the results are clearly presented. The authors make a strong claim about matching the values of the movements. However, the analytical approach to matching them could be more detailed. Currently it is a bit qualitative. Specific suggestions follow.

22) The analysis of the effects of learning on firing rates shows a reasonable correlation. But more sophisticated methods exist for examining such relationships. Specifically, it would be better to fit a simple reinforcement learning model to the data, and correlate value with neural activity. The analysis that correlates number of consecutive correct trials is a bit harder to interpret, although it may be getting at the same point.

23) The RL model would also allow you to more rigorously examine similarities between the reward value representations in the SG and IS blocks. At this point they are only being qualitatively compared.

24) For the analysis of the effects of difficulty in the SG blocks on neural activity, why not just run an ANOVA with difficulty level as a factor? Comparing hard vs. difficult levels is approximately doing this, but since the task design used fixed levels, why not just use them all in the analysis? This could be done in an ANOVA with odor mixture as a factor as well.

25) Also, if an RL model is fit to the learning data, this can be used to estimate accuracy as a function of trials in IS blocks. Accuracy can also be extracted from the psychometric curves in the SG blocks. Accuracy could then be used a factor in an ANOVA analysis of neural activity. How many neurons show a main effect of accuracy? How many show an accuracy by task interaction? Log accuracy should also be analyzed, as this is often more correlated with neural activity in choice tasks than accuracy.

26) A similar study has been carried out in macaques, comparing lateral prefrontal and the caudate nucleus, Seo et al., Neuron, 2012. Which areas do the authors think would be more strongly involved in the SG component of the task?

[Editors' note: further revisions were requested prior to acceptance, as described below.]

Thank you for resubmitting your work entitled "Basal ganglia output reflects internally-specified movements" for further consideration at *eLife*. Your revised article has been favorably evaluated by Timothy Behrens as the Senior editor, a Reviewing editor, and two reviewers.

The manuscript has been improved but there are some remaining issues that need to be addressed before acceptance, as outlined below:

1) One of the reviewers still feels strongly that the main point of analyzing ipsi /contra versus preferred/non preferred direction was not understood. So the reviewer now provides a more detailed explanation of what was the point. We ask the authors to please submit new analyses and details regarding this point (where possible).

2) In addition, it would be good to fix some of the small points raised.

3) Finally, the authors still insist that the data presented are not consistent with a value-biasing view of basal ganglia function. The editors and reviewers felt that the data and results presented focus on the difference between stimulus-driven versus internally-guided movements, which is orthogonal to value biasing (and does not formally exclude value biasing). Therefore, we urge the authors to further revise the manuscript regarding this point.

Reviewer #1:

4) As I mentioned before, I am not arguing that the authors' data are wrong. I have been trying to find a more straightforward way to present the authors' data. My main concern is that the current version may not transmit the basic and essential part of the authors' message to the readers. For this reason, I still have a strong suggestion to use the neuronal activity difference between ipsilateral and contralateral choices, for the following reasons.

Let's start from Figure 3. Does this neuron contribute to the choice? To answer this question, I would set this neuron in SNr on both sides. If the animal decides to choose Rightward movement, the neuron in Right SNr (i.e., Ipsi) is more active than the neuron in Left SNr (i.e., Contra). This would lead to the bias toward Rightward movement (Figure 7). Importantly, the choice bias is stronger in IS than SG (Figure 3). Therefore, the neuron does contribute to the choice using the difference in its activity between Ipsi and Contra choices, and does so more strongly in IS than SG condition.

Checking the neuron's activity for only one choice (i.e., preferred or anti-preferred) would not indicate whether the neuron can contribute to the choice. That's why the data points shown in Figure 3 and Figure 4 provide no clear message. Two data points are associated with one neuron. By looking at these two points in Figure 3, we can guess if the neuron can contribute to the choice; the slope between the two data points indicates the ratio of the directional bias (IS-bias/SG-bias). But, I bet most readers would not reach such detailed understanding. Moreover, the pair of data points is shown for only one example neuron. How about the others?

I understand the authors' logic. The average slope of all data points (regression line) in Figure 3 is higher than 1. Therefore, IS-bias is larger than SG-bias overall. But, the basic information – choice signal carried by individual neurons – is missing. Let me ask a basic question. How many SNr neurons were capable of contributing to the choice? Would the data in Figure 4 answer the question? I doubt it. For example, there are neurons that showed significant correlations with both preferred and anti-preferred directions. Some of them had the same polarities; they would increase (or decrease) their activity during both Ipsi and Contra IS blocks. Such neurons are probably not capable of contributing to the correct choice.

How can we quantify the choice capability of each neuron, then? My only suggestion is to use the difference between Ipsi and Contra directional choices. This exactly matches the model shown in Figure 7.

Other than my critical comments, I do find some of the authors' data interesting. For example, I find two dominant groups of neurons in Figure 4) neurons showing positive correlations with their preferred directions, and 2) neurons showing negative correlations with their anti-preferred directions. This indicates that both groups of neurons enhanced their choice signals, but in different ways: 1) enhancing the dominant signal, and 2) suppressing the non-dominant signal. This reminds me of a basic mechanism of motor control: To change the orientation of a joint, the agonist muscle contracts or the antagonist muscle relaxes, or both contract (or relax) in different amounts. These mechanisms may be useful in different contexts. In this sense, the authors' data may suggest that different groups of SNr neurons contribute to such different mechanisms of joint orientation, in this case, head or eye orientation.

However, this question is one step beyond the main question: Which neurons can contribute to the choice of orientation? To address this main question, the authors should focus on the difference in each neuron's activity between the Ipsi- and Contra-directions, which is similar to the difference in contraction force between agonist and antagonist muscles.

I have some specific questions (below).

5) Data in Figure 3 are different from the ones in the original manuscript. In particular, the two data points for the example neuron are very different. Please give me an explanation.

6) How was the preferred direction defined? I understand that the authors used ROC analysis. Were both SG and IS trials included? Weren't there any neurons that showed a bit different preference between SG and IS trials? For example, the dark blue data point at the far right in Figure 3. This means that the neuron responded strongly in the anti-preferred direction in SG trials. In other words, it responded more strongly in the preferred direction. Which of the dark red data points belongs to this neuron? In any case, this may raise another issue of preferred vs. anti-preferred directions.

Reviewer #2:

7) I have only one brief comment. Otherwise the authors have carefully addressed all of my concerns in detail.

Figure 4 doesn't show value estimates evolving over trials as stated in reply to reviewers and Methods. "In well-behaved IS blocks, *V_dir,t_* tended to asymptotically approach 1 as the mouse consistently returned to the rewarded port (Figure 4)." Was this erroneously not included or with this statement did you just mean you were using the RL algorithm to generate the value estimates and these value estimates were used in Figure 4? It's not clear.

---

## [Author Response]

*There are several issues to be addressed before the manuscript can be considered further. Please find attached below the original comments and also a summary of the most important issues to take into account.*

*1) The reviewers suggest new analyses to quantify the activity of single SNr neurons in the different conditions. They specifically suggest that the authors analyse the activity of individual neurons during ipsi versus contralateral movements.*

We have performed several new analyses to quantify the activity of single SNr neurons:

A) As described in our response to Comment #7, we have reanalyzed neural activity with respect to ipsiversive vs. contraversive movements wherever possible (Revised Figure 5; subsection “Modulation of SNr activity by task-relevant variables”, first paragraph). However, we believe that it remains necessary to segregate trials with respect to the preferred vs. antipreferred direction of the neuron in some cases (Revised Figures 3D and 4; subsections “SNr activity differs for stimulus-guided and internally-specified movements”, second and last paragraphs), as described in our response to Comment #7.

B) As described in our responses to Comments #22 and 25, we have used a reinforcement learning model to provide an estimate of movement value, and calculated how firing rate depends on this estimate over the course of a block of internally-specified (IS) trials (Revised Figure 4, subsection “SNr activity differs for stimulus-guided and internally-specified movements”, last paragraph and subsection “Sign of activity change during delay epoch”).

C) As described in our responses to Comments #11 and 24, we have calculated how neural activity depends on the difficulty of the odor discrimination – or an associated variable, such as uncertainty, or the estimated value of each movement direction – with respect to each mixture ratio separately (Revised Figure 5, subsection “Modulation of SNr activity by task-relevant variables”, first paragraph).

*2) The reviewers ask for new analyses to demonstrate that activity preferentially encodes internally-generate movements versus stimulus-guided throughout different epochs of the movement. In the current version of the study, the activity is measured at constant times in the trial which, given the fact that reaction times are different, may mean different epochs in relation to movement are being compared.*

We believe that this comment is related to Comment #14; please also see our response to it below. However, this comment also raises distinct points, which we address here. We first clarify two points: First, our primary focus is on the delay epoch (from odor valve open to odor port exit; Figure 1), which precedes the movement, rather than epochs during the movement itself. Second, we defined reaction time as the time from go signal to reward port entry. (While it would also be sensible to define reaction time as the time from go signal to odor port exit, we find that exiting the odor port and moving to the reward port are more naturally considered to be part of the same process, so we consider them together.) Since reaction time was generally longer in stimulus-guided (SG) than IS trials (Figure 1), the end of the movement (defined as reward port entry) generally occurred later in SG than IS trials. However, given that our delay epoch ended at the beginning of the movement (odor port exit), and not at the end of the movement, it has the same relationship with movement in both SG and IS trials: it is the period of time immediately preceding movement. We note that this would not be true of other, otherwise reasonable, definitions of the “delay epoch,” e.g., from odor valve open to the go signal; in fact, we have analyzed this epoch as well and found while preparing this resubmission that we had originally reported some results from it. We now report only results from the epoch as defined in Figure 1, resulting in some slightly different numbers in the resubmission, none of which change any of our overall results.

However, we believe that the primary concern raised here, as well as in Comment #14, is that the direction of movement can be selected earlier in IS than SG trials, and that examining activity in the same fixed epoch (the delay epoch: odor valve open to odor port exit) therefore does not provide an appropriate comparison. Instead, perhaps we should consider defining our epoch as having a fixed length but starting relative to when the direction of movement can first be selected; for example, the epoch could begin with odor port entry for IS trials and odor valve open for SG trials (note that, if we simply prepend the period between odor port entry and odor valve open to the delay epoch for IS trials only, we would see an even larger difference between IS and SG trials than we show in Figure 3). We have considered this possibility (among several others), but it would be problematic because many neurons exhibit activity late in our current delay epoch in IS trials, which is likely related to movement selection but would be lost if the epoch ended earlier. Ultimately, we do not believe that there exists a better definition of the relevant epoch to our study than our delay epoch: We are most interested in the activity immediately preceding movement that it captures, because it is at this time that, via modulation of activity in downstream structures, specific movements are facilitated or inhibited. We have clarified this reasoning in the Results section when we first introduce the delay epoch (subsection “SNr activity differs for stimulus-guided and internally-specified movements”, first paragraph), and discuss the relationship between activity preceding stimulus delivery and activity during the delay epoch in the Discussion (second paragraph; also see response to Comment #14).

Finally, we suggest that, rather than being problematic, the fact that activity may start earlier in IS trials captures an important difference between internally-specified and stimulus-guided movements in the real world: the former may be selected earlier than the latter (Introduction, first paragraph). Further, the idea that direction-related activity in IS trials precedes stimulus delivery is consistent with the value-biasing view of basal ganglia (BG) function (Hikosaka et al., 2006), in which reward modulates activity prior to stimulus delivery, resulting in more valuable movements being facilitated relative to less valuable movements (Introduction, first paragraph, Discussion, third paragraph).

*3) The reviewers also comment that SNr-SC connections are largely ipsilateral in different species. The data presented almost suggests that the contralateral connection is equally common. How do the authors explain this?*

As described in our response to Comment #15, we agree that our idea that contraversive-preferring SNr neurons comprise the crossed nigrotectal projection is speculative and remains to be tested – particularly given the fact that about as many SNr neurons exhibit contraversive as ipsiversive preference – and therefore distracts from our main point about how SNr output may promote internally-specified movements. In the revised manuscript, we have therefore removed nearly all of our discussion about the crossed projection, and we have simplified Revised Figure 7 to show only the (more numerous, and more commonly studied) uncrossed nigrotectal neurons in relation to stimulus-guided and internally-specified movements. In addition, we explicitly state that, while contraversive-preferring SNr neurons may project to the contralateral SC, they may also project to non-tectal targets (e.g., thalamus) (Discussion, fifth paragraph).

*4) The reviewers propose other ways of quantifying and analysing/fitting the data; namely using a simple RL model.*

As described in our responses to Comments #22 and 25, in the revised manuscript we use a reinforcement learning model to provide an estimate of movement value (subsection “Sign of activity change during delay epoch”). We then use this estimate to calculate how firing rate evolves over the course of a block of IS trials (Revised Figure 4, subsection “SNr activity differs for stimulus-guided and internally-specified movements”, last paragraph) as well as how it influences firing rate among other factors (subsection “Regression model”).

5) The flow of the manuscript tries to set up an opposition between value-based action selection or movement biasing, versus internally generated action selection or movement biasing. However, the study does not address that, it rather addresses the comparison between similarly valued stimulus-driven versus internally-guided movements (in other formulations feedforward versus feedback driven, sensorimotor versus motor sensory). Therefore, the study does not address that expected value does not influence the activity of SNr during the execution of similar internally-guided movements. It does compare the activity of similarly valued movements, internally-guided are more represented than stimulus-driven. The study is not contrary to the value-biasing view of basal ganglia function, it just shows that basal ganglia cares more about self-paced that stimulus-driven movements (even for the same value). Therefore, the Abstract; Introduction, Discussion, and flow should focus on the difference between stimulus-driven versus internally-guided movements.

We agree that our findings do not contradict the value-biasing view of BG function, and that in order to do so we would have had to vary value across otherwise-identical IS movements and show that SNr activity did not change. Our intent, rather, was to examine whether movements specified by internal representations in general (of which value is one) modulate BG activity. In this sense, our results “extend the model underlying the value-biasing view of BG function (Hikosaka et al., 2006) by suggesting that the influence exerted by the SNr on downstream motor regions is modulated by internal representations beyond value alone” (Discussion). We recognize that this point was unclear in the original submission and have therefore rewritten several passages throughout the revised manuscript:

Revised Abstract: We changed “We found that, contrary to the value-biasing view of basal ganglia function, activity in the substantia nigra pars reticulata, a basal ganglia output, predictably differed preceding internally-specified and stimulus-guided movements” to “We found that activity in the substantia nigra pars reticulata, a basal ganglia output, predictably differed preceding internally-specified and stimulus-guided movements, which is not accounted for by the value-biasing view of basal ganglia function”.

Revised Introduction: We now state, “We therefore asked whether BG activity mediates the influence of value specifically or, more generally, of internal goals on movement selection”; and “However, if the BG are not limited to mediating the influence of value but instead mediate the influence of internal goals in general, their output would differ under these two conditions”.

Revised Results: We changed “However, if value is only one of a number of internal representations contributing to movement selection, then these movements would differentially engage the BG” to “However, if other internal representations beyond value are also integrated by the BG, then stimulus-guided and internally-specified movements may differentially engage the BG despite being equally valuable”.

Revised Discussion: We now state, “We propose that a similar model can also explain how internally-specified movements, *more generally*, are facilitated”.

*Specific Comments:*

*Reviewer #1:*

*6) This study provides data suggesting that the basal ganglia contribute to the internally triggered motor behavior. It has been known that patients with basal ganglia dysfunction may have difficulty in initiating movements internally. For example, patients with Parkinson's disease often have difficulty in initiating movements (e.g., walking) if there is no sensory guidance (Glickstein & Stein, Trends Neurosci 1991) and patients with bilateral pallidal lesions may not initiate goal-directed behavior spontaneously (Laplane et al., J Neurol Neurosurg Psychiatry 1984). However, the underlying mechanism is still unclear. In this sense, this study is important.*

*A difficulty of this theme is how to guide the subject to initiate movements internally. Without any sensory guidance, motor behavior is likely uncontrollable and, if so, behavioral as well as neuronal data may not be feasible for statistical analysis. In such a difficult context, the authors devised a simple but controllable procedure for mice. Thus, this manuscript is valuable.*

*With my high expectation, I now have many suggestions and comments, as shown below.*

It is important to discuss the results of this study in relation to the well-known clinical symptoms described above.

We thank the reviewer for suggesting that we discuss our results in the context of paradoxical kinesia. Indeed, we have done so when discussing this research with colleagues, and we agree that it strengthens the manuscript to do so as well. We have therefore added the following text to the Introduction and Discussion of the revised manuscript:

Revised Introduction: “Notably, it has been proposed that Parkinsonian patients exhibit more bradykinesia when initiating internally-specified than stimulus-guided movements because the latter engage pathways outside of the BG (Glickstein and Stein, 1991). However, whether the BG themselves are differentially engaged by these two types of movements has not been tested”.

Revised Discussion: “Interestingly, patients with Parkinson’s disease and other BG pathologies have been reported to exhibit greater deficits in the initiation of internally-specified than visually-guided movements (Forssberg et al., 1984; Laplane et al., 1984; Azulay et al., 1999). […] Our results suggest that differential processing of internally-specified and visually-guided movements within the BG themselves may also contribute to this clinical observation”.

7) The authors analyzed the preferred and antipreferred directions separately throughout the description of the results, which makes the interpretation very complex and difficult. In the end, to interpret the data as a whole the authors focused on the difference in neuronal activity between the contralateral and ipsilateral choices (Figure 7). Aiming at this goal, it would be much better to use this measure (i.e., Ipsi-Contra) from the beginning. Below are my suggestions.

In the revised manuscript we have changed many of the figures according to the reviewers’ suggestions (Revised Figure 3–Figure 7), and we address specific comments about figures in many of our responses below. Here, we address the general idea of replacing, in Figure 3–Figure 5, our separate analyses of activity in trials in the preferred and antipreferred directions with a measure of the difference in activity between ipsiversive and contraversive trials. As described in detail below (see responses to Comments #9 and 10), we suggest that, in some cases, it makes sense to analyze the data separately for preferred and antipreferred trials.

For example, the main finding in Figure 3 is that the relationship between delay epoch activity in SG and IS trials depends on whether the movement was in the preferred or antipreferred direction of the neuron. For trials in the preferred direction, activity in IS trials was higher than activity in SG trials (Figure 3, red circles). For trials in the antipreferred direction, activity in IS trials was lower than activity in SG trials (Figure 3, blue circles). Figure 8 shows the same data displayed as suggested by the reviewer (difference in activity between ipsiversive and contraversive trials plotted for SG vs. IS trials). This display does not convey the main finding as well as in Figure 3 (see response to Comment #9).

Author response image 1.Alternative display for Revised Figure 3.Difference in delay-epoch firing rate between ipsiversive and contraversive trials in stimulus-guided vs. internally-specified trials in the corresponding session. Black circles show neurons with a significant direction preference (corresponding to black bars in Figure 3; gray circles show neurons without a significant direction preference (corresponding to gray bars in Figure 3. Only correct trials are included; all choices on 50/50 SG trials were considered correct regardless of whether they were rewarded (as in Figure 3). We believe that this display is not as informative as Revised Figure 3 (see response to Comment #9).**DOI:**
http://dx.doi.org/10.7554/eLife.13833.012

Similarly, for Revised Figure 4, it is important that we display correlations separately for trials in the preferred and antipreferred directions, rather than in the ipsiversive and contraversive directions: As we clarify in the revised manuscript, given the results in Figure 3, we would expect to find positive correlations for trials in the preferred direction and negative correlations for trials in the antipreferred direction (subsection “SNr activity differs for stimulus-guided and internally-specified movements”, last paragraph), whereas this pattern would not hold for trials in the ipsiversive and contraversive direction (see response to Comment #10).

For the analysis shown in Revised Figure 5, we could segregate trials either with respect to the direction preference of the neuron or with respect to the direction relative to the recording side. We have therefore revised this figure to show data on ipsiversive and contraversive trials separately, as suggested by the reviewer (see response to Comment #11).

Finally, the functional model shown in Revised Figure 7 (which is now simplified; see response to Comment #15) incorporates both neuronal direction preference and the direction relative to the recording side. As described in the Revised Discussion (fourth paragraph) and the legend, the relative activity of the SNr neurons shown in the model (“relative” with respect to a left vs. right SNr neuron and with respect to an IS vs. an SG movement) is based on whether the direction of movement under consideration (in this case, a rightward movement) is in the preferred or antipreferred direction of the neuron. We therefore believe that the current set of Revised Figure 3–Figure 5, which incorporate many of the reviewers’ suggestions, most clearly convey our main findings.

8) Figure 3: Rearrange graphs in a 2x2 format: SG (left) vs. IS (right), Ipsi (top) vs. Contra (bottom). The Ipsi-Contra difference would become more evident in IS, but not SG.

We thank the reviewer for this suggestion; we agree that the suggested format makes it easier to compare across both trial type and direction. We have made this change in the revised manuscript (Revised Figure 3).

9) Figure 3: In the current version, it is unclear how individual neurons behaved. My suggestion is to plot the Ipsi-Contra difference for each neuron. The distribution of data points would be similar to the current graph. I would then analyze data statistically, including populational trend (e.g., Ipsi-Contra difference is larger in IS than SG) and classification of individual neurons (e.g., IS-preferring, SG-preferring).

Our goal in Figure 3 was to show how the activity of each direction-selective neuron (corresponding to the black bars in Figure 3) compared between IS and SG trials. We plotted activity separately for trials in the preferred (in red) and antipreferred (in blue) direction of the neuron in order to highlight the finding that more neurons: a) exhibited higher activity in IS trials than SG trials in their preferred direction (84/101 vs. 17/101; p = 2.6 x 10^-11^, χ^2^ test), and b) exhibited higher activity in SG trials than IS trials in their antipreferred direction (76/109 vs. 33/109; p = 3.8 x 10^-5^, χ^2^ test). We have clarified this finding in the summarizing sentence of the relevant paragraph in the revised manuscript (subsection “SNr activity differs for stimulus-guided and internally-specified movements”, second paragraph).

In addition, we have plotted the data as suggested by the reviewer (Figure 8). As the reviewer predicted, the difference between firing rates on ipsiversive and contraversive trials tends to be larger on IS than SG trials (i.e., the cloud of points is skewed vertically relative to the x = y line). However, without showing the IS vs. SG data separately for trials in the preferred and antipreferred direction (Figure 8), the main finding of this figure – that more neurons exhibited higher activity in IS trials than SG trials in their preferred direction, and more neurons exhibited higher activity in SG trials than IS trials in their antipreferred direction (as described above) – is not clearly conveyed. We therefore believe that Figure 3 most clearly achieves our intended goal.

10) Figure 4: In the current version, 4A has the same issue as for Figure 1, and 4B is almost meaningless. I would use the Ipsi-Contra difference for 4A for each neuron. Then, show the across-trial changes (perhaps, using a regression line) for a number of neurons; in this case, it may be useful to use the absolute value of the Ipsi-Contra difference, because it would increase across trials. My another suggestion is to plot the Ipsi-Contra difference across time within a trial; the value would increase for IS than SG (as in Figure 3).

In Revised Figure 4, we have now calculated correlation between the firing rate and the value of each port [calculated using a reinforcement learning model, as suggested by Reviewer 2 (see Comment #22), trial by trial (subsection “SNr activity differs for stimulus-guided and internally-specified movements”, last paragraph). We agree with Reviewer 2 that this model provides a more sophisticated method for estimating the extent to which movements are selected based on internal representations, which we expect to increase during the course of an IS block. We believe that this new analysis also addresses the current comment by strengthening Figure 4 in the manner that Reviewer 1 intended. In addition, we have clarified that the results shown in Figure 4 are consistent with the pattern of results shown, separately for trials in the preferred and antipreferred direction, in Figure 3 (in the aforementioned paragraph).

11) Figure 5: The current version includes multiple comparisons, although the graphs do not visually show the results of the comparisons. My suggestion is to present simple histograms showing the absolute values of the Ipsi-Contra difference, separately for IS, SG (easy), and SG (difficult).

Our goal in Figure 5 was to display whether the apparent difference in activity preceding IS and SG movements could be accounted for by a dependence on difficulty, or an associated variable such as uncertainty or the estimated value of each movement direction. In order to do so, we must show a within-neuron comparison between easy and difficult discrimination trials, and also show which neurons exhibited a difference between SG and IS trials. If the same neurons that exhibited a difference between SG and IS trials also showed a dependence on difficulty, it would indicate that the apparent dependence on trial type (SG vs. IS, shown in Figure 3) was actually due to a dependence on difficulty (easy vs. difficult).

We have displayed the data as suggested by the reviewer (Figure 9). As expected given the results shown in Revised Figure 3, the difference in firing rate between ipsiversive and contraversive trials appears to be larger in IS (Figure 9) than in SG trials (Figure 9) However, in this display, the within-neuron comparison is lost, as is the relationship between trial-type- and difficulty-dependence. We therefore believe that this display is not maximally informative.

Author response image 2.Alternative display for Revised Figure 5.Difference in delay-epoch firing rate between ipsiversive and contraversive trials in easy SG trials (**A**), difficult SG trials (**B**), and IS trials (**C**). Neurons with a significant difference in ipsiversive vs. contraversive firing rate are shown in black. We believe that this display is not as informative as Revised Figure 5 (see response to Comment #11).**DOI:**
http://dx.doi.org/10.7554/eLife.13833.013

However, in Revised Figure 5, we have now separated the panels by absolute direction (ipsiversive and contraversive), as suggested by the reviewer (see Comment #7), instead of by the preferred or antipreferred direction of the neuron. In addition, we have made it much easier to see that there was little overlap (purple) between those neurons that exhibit dependence on trial type (red) and those that exhibit a dependence on difficulty (blue). For further clarity, those direction-selective neurons that do not exhibit a dependence on either trial type or difficulty (ipsiversive: n = 94; contraversive: n = 108) are not shown, and we have narrowed the range shown on the x- and y-axes.

12) Figure 6: This figure is too complicated and does not add any support for the conclusion. Moreover, the data include both SG and IS, which I find no benefit. The previous choice should have a larger effect in IS than SG. The current choice should start earlier for IS than SG. My suggestion is to do the same analysis separately for SG and IS, although it may provoke more issues.

As clarified in the revised manuscript, we believe that the regression analysis shown in Figure 6 is valuable because it provides an unbiased method for determining which of several inter-related factors influence neural activity (subsection “Modulation of SNr activity by task-relevant variables”, second paragraph). In addition, we have added text to the Results section to clarify how the results of this analysis contribute to our conclusions (subsection “Modulation of SNr activity by task-relevant variables”, second paragraph). Specifically, we explain that the observed dependence on previous choice is particularly interesting because this variable is critical for determining, in an IS block, which direction is associated with reward (subsection “Modulation of SNr activity by task-relevant variables”, third paragraph). Finally, we have simplified Revised Figure 6 by deleting the line and shading corresponding to a dependence of firing rate on reaction time, which we agree was a distraction from our main conclusion.

Further, we attempted to perform the same analysis as in Revised Figure 6 separately for SG and IS trials, as suggested by the reviewer. However, as the reviewer noted might be the case, there are problems interpreting the results: In IS trials, previous choice and current choice are highly correlated (by design). Therefore, we cannot disambiguate which of these factors activity depends upon.

13) I have to add one issue to my own suggestion – focus on the Ipsi-Contra difference. The feature that is orthogonal to this measure is the average magnitude of the Ipsi & Contra activity. If this is high, SC on both sides would be inhibited and the action (e.g., orienting) would be suppressed in general. If low, the action would be enhanced in general. The authors need to consider if this feature contributes to any difference in animals' choice behavior in general.

This is a very good point. While we show how several factors modulate activity from its baseline level (Figure 3, Figure 5 and Figure 6, and Table 1), the absolute level of activity is important for determining the level of inhibition on downstream motor structures. However, given the context of the decision required by our task (to move left vs. to move right), we believe that the critical factor is the relative difference between activity promoting leftward and rightward movements, as illustrated in Revised Figure 7. We have clarified this point in the revised manuscript (Discussion, fourth paragraph).

*Below, I add my comments (rather than suggestions) which can be critical.*

14) Activity of SNr neurons started differential (ipsi vs. contra) earlier in IS trials (even before odor delivery) than SG trials (basically after odor delivery), as exemplified in Figure 3. Now, the magnitude of neuronal activity was measured during the delay epoch which started with odor delivery. This raised the possibility that the differential activity was higher in IS trials than SG trials simply because the differential activity started earlier in IS trials. Please answer this question.

We agree with the reviewer that direction preference during the delay epoch could be stronger in IS trials than SG trials in part because mice could select their direction movement prior to stimulus delivery on IS, but not SG, trials, allowing direction preference to develop earlier in IS trials. To quantify this, we calculated the difference in the strength of preference between SG and IS trials during the delay epoch and during the epoch from odor port entry and odor valve open (which we refer to here as the “prestimulus” epoch). We indeed found that the larger the difference between SG and IS trials during the prestimulus epoch, the larger the difference during the delay epoch (r = 0.47, p = 4.8 x 10^-18^; Figure 10). These results are consistent with a) the idea that, in the real world, internally-specified movements can be selected earlier than stimulus-guided movements; and b) the value-biasing view of BG function (Hikosaka et al., 2006), in which reward modulates activity prior to stimulus delivery, resulting in more valuable movements being facilitated relative to less valuable movements. In the revised manuscript, we note this correlation analysis and provide the results (Discussion, second paragraph), and we clarify that the value-biasing view also hypothesizes that SNr activity is modulated by reward prior to stimulus presentation (Introduction, first paragraph and Discussion, third paragraph).

Author response image 3.Relationship between activity during prestimulus and delay epochs.Abscissa shows the absolute value of the difference in strength of direction preference between SG and IS trials during the delay epoch, where the sign preserves the relationship between preference in SG and IS trials across epochs (e.g., if pref_SG_ > pref_IS_ in the prestimulus epoch, ordinate shows pref_SG_ – pref_IS_). Black circles show neurons with a significant direction preference (corresponding to black bars in Figure 3; gray circles show neurons without significant direction preference (corresponding to gray bars in Figure 3. See response to Comment #14.**DOI:**
http://dx.doi.org/10.7554/eLife.13833.014

We considered including Figure 10 in the Results section of the revised manuscript itself, but ultimately decided against doing so: Our analyses in Revised Figure 3–Figure 5 and Table 1 currently focus on the delay epoch, and introducing the prestimulus epoch in Revised Figure 3 (where this panel would most naturally fit) would therefore disrupt the flow of the manuscript. Please also see our response to related Comment #2.

15) The different connections shown in Figure 7 may explain the functions of ipsiversive and contraversive SNr neurons, but this is a hopeful scheme with no evidence or hints. Virtually all anatomical studies have shown that the majority of the SNr-SC connections are ipsilateral in various mammals. The contralateral SNr-SC connection is very few in rats (Beckstead et al., J Neurosci 1981), which may be true in mice. The authors found that contraversive SNr neurons (thus, projecting to the contralateral SC) are slightly more common than ipsiversive SNr neurons (thus, projecting to the ipsilateral SC). This raises a concern about the scheme in Figure 7.

We agree that the idea that contraversive-preferring SNr neurons comprise the crossed nigrotectal projection is speculative and remains to be tested, and therefore distracts from our main point about how SNr output may promote internally-specified movements. In the revised manuscript, we have therefore removed nearly all of our discussion about the crossed projection, we have simplified Revised Figure 7 to show only the (more numerous, and more commonly studied) uncrossed nigrotectal neurons in relation to stimulus-guided and internally-specified movements, and we have clarified the description of Figure 7 in the legend. In addition, we explicitly state that, while contraversive-preferring SNr neurons may project to the contralateral SC, they may also project to non-tectal targets (e.g., thalamus) (Discussion, fifth paragraph). Finally, at the end of the Introduction, we clarify that “internally-specified movements may be promoted over stimulus-guided movements by BG activity”.

*Other comments and questions:*

16) Results: "if value is only one of a number of internal representations contributing to movement selection, then these movements would differentially engage the BG." What does this mean?

In the Introduction of the revised manuscript, we clarify our main objective as examining whether “the BG primarily mediate the influence of value, [in which case] their output when selecting equally valuable stimulus-guided and internally-specified movements would be similar. However, if the BG are not limited to mediating the influence of value but instead mediate the influence of internal goals in general, their output would differ under these two conditions”. Likewise, we have rewritten the sentence in question as: “However, if other internal representations beyond value are also integrated by the BG, then stimulus-guided and internally-specified movements may differentially engage the BG despite being equally valuable”.

*17) Results: "We recorded from 298 well isolated left SNr neurons"*

Why did you record SNr neurons only on the left side in 4 mice?

We assume – fairly, we believe – that the relationship between the activity of neurons in each SNr and movement direction is symmetric, such that relative direction (ipsiversive vs. contraversive) is important to activity while absolute direction (left vs. right) is not. Therefore, it is only necessary to record from one SNr. Since it is also more experimentally convenient to do so, this is standard practice in our and other awake behaving recording experiments (e.g., Thompson and Felsen, 2013; Kiani et al., 2014). In our bilateral model (Figure 7), we treat right SNr neurons as “antineurons” to the left SNr neurons that we recorded, in the sense that we assume the same relationship between neural activity and relative direction (ipsiversive or contraversive) (Britten et al., 1992).

*18) Results: "while we found that neurons were equally likely to prefer upcoming ipsiversive and contraversive movements (Figure 3), they were more likely to exhibit a higher firing rate preceding IS movements in their preferred direction and preceding SG movements in their antipreferred direction (Figure 3)."*

It is unclear what this sentence means.

In the revised manuscript, we have clarified that SNr neurons “were more likely to exhibit a higher firing rate preceding IS – as compared to SG – movements in their preferred direction, and preceding SG – as compared to IS – movements in their antipreferred direction (Figure 3)” (Results).

*19) Results: "We observed no difference in activity between SG trials in which different mixtures were presented but movement direction was held constant"*

Please show statistical evidence.

Although we have deleted this particular sentence from the revised manuscript, we have addressed this idea in the revised manuscript. We performed an ANOVA to examine whether activity depended on mixture ratio separately for each direction, and found that the activity of some individual neurons depended on mixture ratio (or difficulty or an associated variable) (ipsiversive direction: 39/216 neurons; contraversive direction: 31/216 neurons, p < 0.05, 1-way ANOVA across mixture ratios, Revised Figure 5; subsection “Modulation of SNr activity by task-relevant variables”, first paragraph). Please also see our response to Comment #24.

20) Results: "The difficulty of this decision could, in principle, affect SNr activity." Another possibility is 'uncertainty', rather than difficulty.

In the revised manuscript, we have edited this sentence by replacing “choice confidence” with “uncertainty;” it now reads, “The difficulty of this decision – or an associated variable, such as uncertainty, or the estimated value of each movement direction – could, in principle, affect SNr activity […]”.

21) Results: "there was little overlap between this small population of difficulty-dependent neurons and those neurons that exhibited a dependence on trial type (Figure 5)". This notion is unclear simply by looking at these graphs.

See response to comment #11.

*Reviewer #2:*

*The authors present results from a study in which they recorded neural activity in the SNr while mice carried out a task in which they chose left or right reward ports either on the basis of an odor mixture (stimulus guided) or on the basis of previously rewarded outcomes (internally-guided). They examined neural activity, mostly during the choice period and found that SNr neurons coded ipsiversive and contraversive movements with about equal frequency. Neurons responded more strongly for internally-guided movements in their preferred direction, and more strongly for stimulus guided movements in their anti-preferred direction. They also correlated with learning in the IS blocks.*

*This study addresses an interesting question about the role of the basal ganglia, assessed from the perspective of an output nucleus, in internally generated vs. externally cued movements. Overall the study was well carried out. The dataset is reasonable and the results are clearly presented. The authors make a strong claim about matching the values of the movements. However, the analytical approach to matching them could be more detailed. Currently it is a bit qualitative. Specific suggestions follow.*

*22) The analysis of the effects of learning on firing rates shows a reasonable correlation. But more sophisticated methods exist for examining such relationships. Specifically, it would be better to fit a simple reinforcement learning model to the data, and correlate value with neural activity. The analysis that correlates number of consecutive correct trials is a bit harder to interpret, although it may be getting at the same point.*

This is an excellent suggestion. We have now used a reinforcement learning model to estimate the value associated with movement in each direction on each trial of the task (subsection “SNr activity differs for stimulus-guided and internally-specified movements”, last paragraph and subsection “Sign of activity change during delay epoch”). Briefly, we iteratively updated the value associated with each direction in each trial as Vdir, t=Vdir, t−1+α(Rdir,t−1−Vdir,t−1), where Rdir,t−1is the reward for the given direction in the previous trial in which that direction was chosen (0 for unrewarded and 1 for rewarded) and α is the learning rate (set to 0.1; similar results were obtained with a range of values) (Sutton and Barto, 1998). Since we calculated Vdir, t in each trial of the session (including SG and IS trials), it tended to start near 0.5 (but was not exactly 0.5) at the beginning of each IS block. In well-behaved IS blocks, Vdir, t tended to asymptotically approach 1 as the mouse consistently returned to the rewarded port (Figure 4). Vdir, t was calculated separately for the ipsiversive and contraversive directions and was only updated in trials in which that direction was selected.

In Revised Figure 4, we have replaced “consecutive correct trials” with this estimate of value in each IS trial, which provides a more rigorous analysis and also strengthens our results: 77/157 neurons exhibited a significant correlation (p < 0.05) between firing rate and the number of consecutive correct trials for either direction [Revised Figure 4/77 for trials in the preferred direction (red circles), 29/77 for trials in the antipreferred direction (blue circles), and 13/77 for trials in both directions (purple circles)], with more positive correlations for trials in the preferred direction (p = 2.4 x 10^-6^, χ^2^ test) and negative correlations for trials in the antipreferred direction (p = 5.9 x 10^-6^, χ^2^ test), as we would expect given the pattern of results shown in Figure 3 (subsection “SNr activity differs for stimulus-guided and internally-specified movements”, last paragraph).

23) The RL model would also allow you to more rigorously examine similarities between the reward value representations in the SG and IS blocks. At this point they are only being qualitatively compared.

This is a fair point, and important to address on its own. While we designed the task such that correctly performed trials were equally rewarded, and therefore valuable (Abstract, Introduction, last paragraph), we could not entirely predict and/or control for the animals’ choice behavior, and therefore the value of different trial types that was actually experienced is not necessarily identical. We now clarify this point in the revised manuscript (Discussion, first paragraph). Indeed, when we compared the average value of IS and SG trials (estimated in each IS trial with the reinforcement learning model and in each SG trials using the block-by-block psychometric functions, as suggested in Comment #25) within each mouse, we found that, for 3 mice IS trials were slightly more likely to be rewarded and for 1 mouse SG trials were slightly more likely to be rewarded (although the timescale over which value for each mixture ratio should best be estimated – by trial, block, session, or lifetime – is an open – and interesting – question). However, in Revised Figure 5 we directly examined whether the difference we observed between IS and SG trials in Revised Figure 3 could be due to a dependence on the value of the movement. Specifically, as suggested by the reviewer (see response to Comment #24), we performed an ANOVA to examine whether activity depended on mixture ratio. Given that mice performed at different levels of accuracy on different mixture ratios (Figure 3), and we used accuracy as a proxy for movement value in the RL model, this analysis addresses whether activity is influenced by value. We found that, while the activity of some neurons was influenced by value, this dependence did not account for the observed difference between IS and SG trials (Revised Figure 5;subsection “Modulation of SNr activity by task-relevant variables”, first paragraph).

*24) For the analysis of the effects of difficulty in the SG blocks on neural activity, why not just run an ANOVA with difficulty level as a factor? Comparing hard vs. difficult levels is approximately doing this, but since the task design used fixed levels, why not just use them all in the analysis? This could be done in an ANOVA with odor mixture as a factor as well*.

As suggested, we performed an ANOVA to examine whether activity depended on the mixture ratio presented in SG trials, separately for each direction. Mixture ratio is a useful variable that can serve as a proxy for several other quantities, including discrimination difficulty, uncertainty, accuracy, and the value associated with each side. For example, compared to a 60/40 mixture ratio, a 95/5 mixture ratio is associated with decreased discrimination difficulty, decreased uncertainty, increased accuracy, and increased value at the right port (corresponding to the increased accuracy and therefore higher likelihood of reward, as noted in Comment #25). Therefore, this analysis allows us to examine the influence of several potential factors on firing rate.

For ipsiversive movements, we found that the activity of 39 of the 216 direction-selective neurons depended on difficulty (p < 0.05). However, consistent with the results shown in Figure 5, difficulty dependence was unlikely to explain the dependence on trial type (SG vs. IS): The activity of 21/109 trial-type-selective neurons was difficulty-dependent; this ratio did not differ from the 18/107 non-trial-type-selective neurons that exhibited difficulty dependence (p = 0.32, χ^2^ test). Similarly, for contraversive movements, we found that the activity of 16/101 trial-type-selective neurons was difficulty-dependent; this ratio did not differ from the 15/115 non-trial-type-selective neurons that exhibited difficulty dependence (p = 0.56, χ^2^ test). Given that the value associated with a given side is determined by mixture ratio, these results further support the idea that the dependence of activity on trial type is not due to a dependence on value (see response to Comment #23). We have added the results of the analysis to the revised manuscript (subsection “Modulation of SNr activity by task-relevant variables”, first paragraph).

25) Also, if an RL model is fit to the learning data, this can be used to estimate accuracy as a function of trials in IS blocks. Accuracy can also be extracted from the psychometric curves in the SG blocks. Accuracy could then be used a factor in an ANOVA analysis of neural activity. How many neurons show a main effect of accuracy? How many show an accuracy by task interaction? Log accuracy should also be analyzed, as this is often more correlated with neural activity in choice tasks than accuracy.

As the reviewer suggests, we calculated value (i.e., accuracy; see response to Comment #24) in each IS trial using the reinforcement learning model (see response to Comment #23) and in each SG trial using the psychometric functions. In order to determine the influence of value on firing rate, rather than using an ANOVA – which we believe is not well suited to this analysis because value is a continuous variable, ranging from 0 to 1 for both IS and SG trials – we included a term for value (or log value) and the interaction between value (or log value) and trial type in our regression analysis (subsection “Regression model”). Of the 296 neurons in our population, we found that the firing rate of 108 neurons was influenced by log value and the firing rate of 49 neurons was influenced by the interaction between log value and trial type (some were influenced by both); the firing rate of 105 neurons remained dependent on trial type. Results were similar when we used value itself as a factor instead. These results are as we might expect, given the known dependence of movement-related SNr activity on the value of the movement (Bryden et al., 2011; Sato and Hikosaka, 2002). We have included this information in the revised manuscript (in the aforementioned subsection).

We believe, however, that incorporating these results into Revised Figure 6 itself would detract from the effectiveness of this panel. One of our goals in this panel is to present the extent to which particular variables that, by design, correlate with trial type (previous choice and reaction time) can account for the apparent dependence of firing rate on trial type (Figure 3). It could have been the case, for example, that the activity of very few neurons depends on trial type when these other variables are controlled for—but this is not what we observed (Revised Figure 6). Since we have already shown directly that the dependence of firing rate on trial type is not due to a dependence on value (Revised Figure 5), including value in Revised Figure 6 would be somewhat redundant. Further, incorporating additional factors into this panel would be nontrivial, requiring a 5-set Venn diagram (since we cannot eliminate any of the original factors) and further complicating an already complex panel (see Comment #12).

26) A similar study has been carried out in macaques, comparing lateral prefrontal and the caudate nucleus, Seo et al., Neuron, 2012. Which areas do the authors think would be more strongly involved in the SG component of the task?

We thank the reviewer for calling attention to the study by Seo et al. (2012), in which a similar behavioral paradigm (with “fixed” and “random” blocks of trials) was used to examine representations of action selection and action value in the lateral prefrontal cortex and dorsal striatum. In the revised manuscript, we now contextualize our behavioral paradigm with reference to this paper, as well as other recent studies that employed variants of “fixed-choice” and “free-choice” trials (Pastor-Bernier and Cisek, 2011; Seo et al., 2012; Ito and Doya, 2015; Introduction, last paragraph).

While Seo et al. (2012) found that lateral prefrontal cortex and dorsal striatum were each engaged by both the fixed and random trials, they noted some interesting differences between the regions: the selected action was more robustly represented in lateral prefrontal cortex, and action value was represented in dorsal striatum. It would be interesting to record from these regions (among several others!) in our task. We speculate that dorsal striatum would be more strongly engaged by our IS trials, in which action values need to be re-learned based on choice and reward history, than our SG trials (although it would continue to represent action values during SG trials). Perhaps lateral prefrontal cortex would be more strongly engaged by our SG than IS trials, since selected movements in random trials (which are somewhat analogous to our SG trials) were represented earlier in lateral prefrontal cortex than dorsal striatum.

[Editors' note: further revisions were requested prior to acceptance, as described below.]

*The manuscript has been improved but there are some remaining issues that need to be addressed before acceptance, as outlined below:*

*1) One of the reviewers still feels strongly that the main point of analyzing ipsi /contra versus preferred/non preferred direction was not understood. So the reviewer now provides a more detailed explanation of what was the point. We ask the authors to please submit new analyses and details regarding this point (where possible).*

We thank Reviewer #1 for the further explanation on this point. We have performed new analyses and revised Figure 3 and Figure 4 in order to address the comments; please see our response to Comment #4.

2) In addition, it would be good to fix some of the small points raised.

We are happy to address all of the concerns raised, as described in our responses to Comments #5-7, and we have revised the manuscript accordingly.

3) Finally, the authors still insist that the data presented are not consistent with a value-biasing view of basal ganglia function. The editors and reviewers felt that the data and results presented focus on the difference between stimulus-driven versus internally-guided movements, which is orthogonal to value biasing (and does not formally exclude value biasing). Therefore, we urge the authors to further revise the manuscript regarding this point.

We agree that our results focus on how substantia nigra pars reticulata (SNr) activity differs between stimulus-guided and internally-specified movements, and are in no way inconsistent with the value-biasing view of basal ganglia function. As listed below, we have edited several portions of the manuscript to further clarify this point; we bold the main changes here for clarity. We believe these changes, in addition to the changes in the previous submission, clarify that our results do not argue against – but rather extend – the value-biasing hypothesis. Please note that in the Introduction of both the previous and current submission, we have retained our description of the value-biasing view because it provides appropriate background on how the BG have primarily been studied with respect to modulatory influences on movement selection, which we build on here.

Revised Abstract: We changed “Here, we examine whether value specifically, or internal goals more generally, influence movements via the basal ganglia” to “Here, we examine whether other internal goals, in addition to value, alsoinfluence movements via the basal ganglia”. In addition, we deleted the clause “which is not accounted for by the value-biasing view of basal ganglia function” from the end of the following sentence: “We found that activity in the substantia nigra pars reticulata, a basal ganglia output, predictably differed preceding internally- specified and stimulus-guided movements”.

Revised Introduction: Original text: “We therefore asked whether BG activity mediates the influence of value specifically or, more generally, of internal goals on movement selection. […] However, if the BG are not limited to mediating the influence of value but instead mediate the influence of internal goals in general, their output would differ under these two conditions.” Revised text: “We therefore asked whether BG activity mediates the influence of internal goals, in addition to value, on movement selection. We reasoned that, if this were the case, BG output would differ when selecting equally valuable stimulus-guided and internally-specified movements.”. In addition, we deleted the clause “beyond value” from the end of the sentence “We found that SNr activity predictably differed between these two conditions, supporting the idea that the BG mediate the influence on movement selection of internal goals”.

Revised results: We changed “According to the value-biasing view of BG function, equally valuable stimulus-guided and internally-specified movements would similarly engage the BG. However, if other internal representations beyond value are also integrated by the BG, then stimulus-guided and internally-specified movements may differentially engage the BG despite being equally valuable. In the latter case…” to “If the BG integrates not only value but also other internal representations, then stimulus-guided and internally-specified movements may differentially engage the BG despite being equally valuable. In this case […]”. We recognize that the previous version made it seem as if the predicted results would refute the value- biasing view, which was not our intent.

Revised Discussion: We changed “Our results therefore extend the model underlying the value- biasing view of BG function (Hikosaka et al., 2006) by suggesting that the influence of the SNr on downstream motor regions is modulated by internal representations beyond value alone” to “Our results therefore extend the model underlying the value-biasing view of BG function (Hikosaka et al., 2006) by suggesting that the influence of the SNr on downstream motor regions is modulated by internal representations in addition to value”.

*Reviewer #1:*

*4) As I mentioned before, I am not arguing that the authors' data are wrong. I have been trying to find a more straightforward way to present the authors' data. My main concern is that the current version may not transmit the basic and essential part of the authors' message to the readers. For this reason, I still have a strong suggestion to use the neuronal activity difference between ipsilateral and contralateral choices, for the following reasons.*

*Let's start from Figure 3. Does this neuron contribute to the choice? To answer this question, I would set this neuron in SNr on both sides. If the animal decides to choose Rightward movement, the neuron in Right SNr (i.e., Ipsi) is more active than the neuron in Left SNr (i.e., Contra). This would lead to the bias toward Rightward movement (Figure 7). Importantly, the choice bias is stronger in IS than SG (Figure 3). Therefore, the neuron does contribute to the choice using the difference in its activity between Ipsi and Contra choices, and does so more strongly in IS than SG condition.*

*Checking the neuron's activity for only one choice (i.e., preferred or anti-preferred) would not indicate whether the neuron can contribute to the choice. That's why the data points shown in Figure 3 and Figure 4 provide no clear message. Two data points are associated with one neuron. By looking at these two points in Figure 3, we can guess if the neuron can contribute to the choice; the slope between the two data points indicates the ratio of the directional bias (IS-bias/SG-bias). But, I bet most readers would not reach such detailed understanding. Moreover, the pair of data points is shown for only one example neuron. How about the others?*

*I understand the authors' logic. The average slope of all data points (regression line) in Figure 3 is higher than 1. Therefore, IS-bias is larger than SG-bias overall. But, the basic information – choice signal carried by individual neurons – is missing. Let me ask a basic question. How many SNr neurons were capable of contributing to the choice? Would the data in Figure 4 answer the question? I doubt it. For example, there are neurons that showed significant correlations with both preferred and anti-preferred directions. Some of them had the same polarities; they would increase (or decrease) their activity during both Ipsi and Contra IS blocks. Such neurons are probably not capable of contributing to the correct choice.*

How can we quantify the choice capability of each neuron, then? My only suggestion is to use the difference between Ipsi and Contra directional choices. This exactly matches the model shown in Figure 7.

We thank the reviewer for further clarifying this concern, which we believe we now address. We entirely agree that the difference in firing rate between ipsiversive and contraversive trials is critical. In the revised manuscript we now display the data as recommended by the reviewer, in two separate panels depending on the direction preference of the neuron (calculated during SG and IS trials combined; see response to Comment #6; Revised Figure 3; subsection “SNr activity differs for stimulus-guided and internally-specified movements”, second paragraph). This way, it is clear that for ipsiversive-preferring neurons, (activity in ipsiversive trials – activity in contraversive trials) is greater for internally-specified (IS) than stimulus-guided (SG) trials, and for contraversive-preferring neurons, (activity in contraversive trials – activity in ipsiversive trials) is also greater for IS than SG trials. In other words, the difference in firing rate between trials in the preferred and antipreferred directions tends to be higher for IS trials than SG trials. We believe that this format most clearly conveys the difference in firing rate between ipsiversive and contraversive trials across our population of neurons. (When ipsiversive- and contraversive- preferring neurons are combined in one panel, as was the case in Figure 8 of the previous submission, the main point is less clear.) For consistency with Revised Figure 3, we have also separated ipsiversive- and contraversive-preferring neurons in Revised Figure 4.

We note that, although we no longer include the previous version of Figure 3, we retain a similar analysis in order to classify neurons as trial-type dependent (i.e., how activity differs between IS and SG trials): We compare firing rates between SG and IS trials separately for the preferred and antipreferred direction of the neuron (in the aforementioned paragraph). We use this classification for the analyses shown in Revised Figure 4 and Figure 5.

Finally, we agree that the connection between Revised Figure 3 and our model (Figure 7) is now more straightforward, and we have significantly simplified the relevant text accordingly (Discussion, fourth paragraph and fifth paragraphs).

*Other than my critical comments, I do find some of the authors' data interesting. For example, I find two dominant groups of neurons in Figure 4) neurons showing positive correlations with their preferred directions, and 2) neurons showing negative correlations with their anti-preferred directions. This indicates that both groups of neurons enhanced their choice signals, but in different ways: 1) enhancing the dominant signal, and 2) suppressing the non-dominant signal. This reminds me of a basic mechanism of motor control: To change the orientation of a joint, the agonist muscle contracts or the antagonist muscle relaxes, or both contract (or relax) in different amounts. These mechanisms may be useful in different contexts. In this sense, the authors' data may suggest that different groups of SNr neurons contribute to such different mechanisms of joint orientation, in this case, head or eye orientation.*

*However, this question is one step beyond the main question: Which neurons can contribute to the choice of orientation? To address this main question, the authors should focus on the difference in each neuron's activity between the Ipsi- and Contra-directions, which is similar to the difference in contraction force between agonist and antagonist muscles.*

*I have some specific questions (below).*

5) Data in Figure 3 are different from the ones in the original manuscript. In particular, the two data points for the example neuron are very different. Please give me an explanation.

As noted in our response to Comment #2 on our original submission, in the original Figure 3 we had reported results calculated with a slightly different definition of the delay epoch – from odor valve open to the go signal – than the epoch shown in Figure 1 (odor valve open to odor port exit). While this alternative definition is not unreasonable, we believe that it is important for the delay epoch to end at the beginning of movement initiation (i.e., odor port exit), so that it has the same temporal relationship with movement across SG and IS trials (see response to Comment #2 on our original submission for further details). In the first revision, we made sure to report only results from the epoch as defined in Figure 1, resulting in some slightly different values than in the original submission, none of which changed any of our overall results. In the current revision we have significantly modified Revised Figure 3 according to Reviewer #1’s suggestion (see response to Comment #4).

6) How was the preferred direction defined? I understand that the authors used ROC analysis. Were both SG and IS trials included? Weren't there any neurons that showed a bit different preference between SG and IS trials? For example, the dark blue data point at the far right in Figure 3. This means that the neuron responded strongly in the anti-preferred direction in SG trials. In other words, it responded more strongly in the preferred direction. Which of the dark red data points belongs to this neuron? In any case, this may raise another issue of preferred vs. anti-preferred directions.

We now clarify in the revised manuscript that we calculated direction preference (shown in Figure 3) based on activity during SG and IS trials combined (subsection “SNr activity differs for stimulus-guided and internally-specified movements”, first paragraph). Indeed, consistent with the results shown in Revised Figure 3, direction preference was generally stronger in IS than SG trials, although preferences calculated in SG trials were highly correlated with those calculated in IS trials (see Figure 11).

Author response image 4.Direction preference calculated in SG vs.IS trials in the same session. Each circle corresponds to one neuron.**DOI:**
http://dx.doi.org/10.7554/eLife.13833.015

*Reviewer #2:*

*7) I have only one brief comment. Otherwise the authors have carefully addressed all of my concerns in detail.*

*Figure 4 doesn't show value estimates evolving over trials as stated in reply to reviewers and Methods. "In well-behaved IS blocks, V_dir,t_ tended to asymptotically approach 1 as the mouse consistently returned to the rewarded port (Figure 4)." Was this erroneously not included or with this statement did you just mean you were using the RL algorithm to generate the value estimates and these value estimates were used in Figure 4? It's not clear.*

We intended the latter: We used the RL algorithm to generate the value estimates and these value estimates are what are shown on the x-axis of Figure 4. It is true that *V_dir,t_*tended to asymptotically approach 1 as the mouse consistently returned to the rewarded port, as would be expected, but in Figure 4 we do not show *V_dir,t_*as a function of trial number; rather, we show firing rate as a function of *V_dir,t_*. In the quoted sentence of the Materials and methods section of the revised manuscript, we have deleted the reference to Figure 4 (subsection “Reinforcement learning model”). In addition, to clarify what we are showing in Figure 4, its legend in the revised manuscript now states that *V_dir,t_*is plotted on the x-axis. We retain the current x axis label within Figure 4 itself (“Extent that choice is internally specified”), as we believe it is most descriptive of the point this figure is trying to convey.